# Removal of Hexavalent Chromium from Aqueous Solutions Using Sulfonated Peat

**Haiqing Li [†], Rongrong Hou [†], Yuefang Chen * and Huilun Chen ***

Beijing Key Laboratory of Resource-Oriented Treatment of Industrial Pollutants, School of Energy and Environmental Engineering, University of Science and Technology Beijing, 30 Xueyuan Road, Haidian District, Beijing 100083, China; lihaiqing0810@163.com (H.L.); hourongrong1994@163.com (R.H.)
* Correspondence: chenyuefang@ustb.edu.cn (Y.C.); chenhuilun@ustb.edu.cn (H.C.)
† These authors contributed equally to this work and should be considered as the co-first authors.

**Abstract:** Peat, a loose and porous material, contains rich organic matter and can be used as an adsorbent. In this study, it is chemically modified by adding sulfuric acid under different conditions, with the aim of producing a modified peat with optimized Cr(VI) adsorption capability. The modified peat exhibited a higher adsorption efficiency than the natural peat throughout the adsorption experiments. The adsorption of Cr(VI) from aqueous solutions correlates with the pseudo-second order kinetic model. In addition, the Langmuir model indicated a maximum loading capacity approximately of 105.4 mg/g, which is a markedly high value compared to some other reported adsorbents. The present study performed single factor experiments and the results indicated that higher temperature conditions result in better adsorption capability, whilst an increase in the pH played a contrary role. According to the orthogonal tests, the pH had the greatest impact on adsorption. The obtained results indicated that sulfonated peat can be effectively applied in removing Cr (VI).

**Keywords:** sulfonated peat; Cr (VI) adsorption; kinetics; equilibrium; orthogonal test

## 1. Introduction

Heavy metal pollution has become an increasingly serious issue in developing economies. Of its various sources, chrome residues are one of the most important given that chromium has long been used in a wide range of industrial activities from electroplating and chromic salt manufacturing to printing, dyeing, and leather tanning, all of which discharge various quantities of chromium effluent [1]. Additionally, dumped chrome residues find their way into soils and water above and below the ground with the help of rain water. Cr(III) and Cr(VI) are characterized as toxic, especially Cr(VI), and have a strong oxidative effect on living cells and mutagenic effects on animals and plants [2]. Cr(VI) can cause a series of lesions once ingested [3]. As such, chromium has been recognized as one of 47 of the most dangerous wastes in the world. In addition, due to the effects of bioaccumulation and biological amplification, chromium pollution poses an increasingly chronic threat to human health [4]. As such, increased attention has been drawn to the damage it causes and how to effectively remove Cr(VI).

Many authors have reported on the ability of different methods to remove Cr(VI) including adsorption, electrocoagulation, ion exchange, chemical precipitation, membrane separation, reduction, and biosorption methods [5–7]. Adsorption is considered as a simple, highly efficient [8], and low-cost method [9]. As such, many studies have been performed on adsorbents used in sewage treatments [10–16]. However, the practical application of many adsorbents such as activated carbon and other nanomaterials are limited due to their high cost; therefore, recent studies have focused on developing new low-cost and high-efficiency adsorbents. Compared to conventional adsorbents for the removal of toxic metals from industrial effluents, peat has many advantages including its low

operating cost and efficient adsorption ability. Peat is a type of organic-inorganic complex formed by the mixture of dead plant and clay minerals under high humidity and airtight conditions. It contains a large quantity of cellulose, hemicellulose, lignin, and humic substances, has a large specific surface area, and many active functional groups such as carboxyl, phenolic hydroxyl, and alcohol hydroxyl groups [17,18]. This enables the biological materials made from peat to remove certain pollutants from wastewater by adsorption, filtration, ion exchange, and complexation [19,20].

Studies on peat as an adsorbent in wastewater treatments can be traced back to the 1970s. Previous studies on the application of peat show that peat is able to remove Ni from wastewater by filtration [21–23]. Hence, this material has been widely used to remove heavy metals from wastewater in slaughterhouses, septic tanks, and the like. In the last decade, a growing number of scientists and researchers have been examining peat as a heavy metal adsorbent, with many of their research projects focusing on the treatment of water contaminated with Cu [24,25], Cd [26], Zn [26], Co [27], Pb [28], or compound heavy metals [26,27,29–31] by filtration. Likewise, many studies on chromium adsorption have used all types of natural peat as an adsorbent [32]. Peat and coconut fiber have also been used to adsorb $Cr^{3+}$ (with an adsorption capacity of 18.75 mg/g) and $Cr^{6+}$ (with an adsorption capacity of 8.02 mg/g) [33]. Previous flow-through column experiments have indicated that an optimal adsorption capacity of 65.87 mg/g can be obtained at a pH of 2.0 [34]. Although unmodified peat can be cost-efficient, the organic matter in peat itself has the potential to enter the water when it is used to adsorb heavy metals.

Various low-cost chemical treatments have been considered to modify the adsorption properties of peat. However, the Cr adsorption capacity of a sample peat modified at a high temperature (200 °C) may remain as low as 0.120 mg/g [35], followed by 4.90 ± 0.01 mg/g in peat modified with NaOH and HCl [36] and 18.6 mg/g in peat modified with NaCl [37]. The observed adsorption efficiency falls below the desired effect. In addition, at present, many researchers use sewage that exhibits low chromium concentrations as their experimental sample, which differs from authentic sewage discharged from factories during real operations as those have high chromium concentrations [36]. For this reason, further research is being focused on peat-based technologies in wastewater treatment applications.

In this study, sulfonation modification was selected as it significantly improves the ion exchange capability of the materials. The present study characterized sulfonated peat for Cr(VI) removal by using scanning electron microscopy (SEM), Fourier transform infrared spectroscopy (FT-IR), zeta potential analysis, and Brunauer–Emmett–Teller (BET) measurements. In addition, the adsorption kinetics and the interaction mechanism of Cr(VI) with the sulfonated peat were examined in detail.

## 2. Materials and Methods

### 2.1. Materials

Natural peat, provided by the Sun Gro Horticulture Company from Canada, was air dried and ground to pass through a 2 cm sieve before being added to concentrated sulfuric acid (95%–98%) at a solid-to-liquid ratio of 1:4 (100 g peat in 400 mL sulfuric acid). Concentrated sulfuric acid was provided by the Jinruilin Technology Company from China. Three different treatments were applied: the peat in the first treatment was steeped in $H_2SO_4$ for 24 h at room temperature, the peat in the second treatment was steeped in $H_2SO_4$ and heated for 2 h in a water bath at 90 °C then steeped 24 h at room temperature, and in the third treatment, the peat was steeped in $H_2SO_4$ and heated for 2 h in an electric cooker at 600 °C then steeped for 24 h at room temperature. The peat modified through these methods was numbered as 1, 2, and 3. Acid liquors were then poured out and the peat was washed using tap water until the pH of the supernatants reached a level between 6 to 7. After that, the three materials were dried at 105 °C before being stored at room temperature.

The Cr(VI) adsorption capacity of the three kinds of modified peat (modified peat 1, modified peat 2, modified peat 3) was then tested in order to identify the best performing one. Three stock solutions of Cr(VI) were prepared by dissolving appropriate amounts of potassium dichromate ($K_2Cr_2O_7$) in

500 mL of deionized water. Potassium dichromate was provided by the Jinruilin Technology Company from China. The three solutions, with concentrations of 50, 100, and 200 mg/L, respectively, were later separately mixed with 0.25 g of the prepared materials in centrifuge tubes and then oscillated for 4 h at room temperature at a speed of 300 rpm in a THZ oscillator. The mixtures were then centrifuged for 5 min before their supernatants were filtered through a 0.45 μm filtration membrane to measure the concentration of the remaining chromium ions.

### 2.2. Characterization of Natural Peat and Sulfonate Peat

The physical changes in the peat were analyzed using a scanning electron microscope (Quanta FEG250, FEI, Hillsboro, OR, USA). The morphological changes were analyzed together with their ability to promote adsorption. The specific surface area and pore diameter of the peat before and after modification were measured using a specific surface area analyzer (TriStar II Plus 2.02, Micromeritics, Norcross, GA, USA), and the zeta potential was analyzed with a Zeta analyzer (Malvern Zetasizer Nano ZS 90, Malvern Panalytical Ltd., Malvern, UK). To define the characteristic chemical bonds present in the sorbent, the materials were subjected to Fourier transform infrared spectroscopy (Nicolet iS50, Thermo Fisher Scientific, Waltham, MA, USA) before and after the adsorption process. This study also analyzed the chromium adsorption process of the modified peat by comparing changes in its specific surface area and pore diameter with those of the natural peat. The adsorption mechanisms of the modified peat were examined using FT-IR by analyzing changes in the functional groups during the entire adsorption process.

### 2.3. Adsorption Isotherms and Their Fitting

Sulfonated peat samples ($0.03 \pm 0.002$ g) were added to 25 mL of 20, 40, 60, 80, 100, 150, 200, 250, 300, 350, 400, and 500 mg/L Cr(VI) solutions and oscillated for 4 h at a rate of 300 rpm under an ambient temperature of about $25 \pm 1$ °C and pH of about 6–7. The material in each tube was then centrifuged and the chromium ion concentrations of the solutions ($C_{eq}$) at equilibrium were measured.

The adsorbed amount served as a metric for the concentration change between the initial solution and the measured filtrate. The adsorbed amount (q, mg/g) was calculated using Equation 1 below [36].

$$q = \frac{V(C_0 - C_{eq})}{W} \tag{1}$$

where $C_0$ is defined as the concentration of Cr(VI) in the initial solution (mg/L), $C_{eq}$ is the concentration of Cr(VI) in the solution at equilibrium (mg/L), $V$ is the volume of the aqueous solution in the centrifuge tube (L), and $W$ is the mass of peat (g).

The concentration of Cr(VI) was determined using an inductively coupled plasma optical emission spectrometer (iCAP 7000, Thermo Fisher Scientific, Waltham, MA, USA) [38].

### 2.4. Adsorption Kinetics and Their Fitting

Peat samples ($0.08 \pm 0.002$ g) were added to 25 mL of 100 mg/L Cr(VI) solutions and shaken at the rate of 300 rpm at room temperature of $25 \pm 1$ °C and pH of about 6–7 for different periods of 10, 20, 30, 40, 50, 60, 70, 80, 90, 120, 150, 180, 210, 240, 270, and 300 min, respectively. The material in each tube was then centrifuged, and chromium ion concentrations of their supernatants were measured by inductively coupled plasma optical emission spectrometry (ICP-OES).

Lastly, the adsorbed amounts for the samples were calculated and the kinetics equations were fitted. To study the adsorption behavior, three kinetics models were used: the pseudo-first order kinetic model, the pseudo-second order kinetic model, and the Elovich model.

*2.5. Factors Influencing Adsorption and the Orthogonal Experiments*

This study analyzed different influencing factors for the removal of Cr, including the pH, peat dose, and temperature. The 50 mg/L Cr(VI) solutions were partitioned into 25 mL portions with different pH and were transferred separately into centrifuge tubes with 0.08 g of modified peat in each tube. The pH of the solutions were adjusted to the following values using an appropriate amount of 0.5 mol/L HNO$_3$ or 0.5 mol/L NaOH: 1.5, 2, 3, 4, 5, 6, 7, 8, 9, 10, and 11. A control sample (pH of 5.13) was prepared by adding Cr without the addition of HNO$_3$ and NaOH. The samples were shaken at the rate of 300 rpm at room temperature of 25 ± 1 °C. Similarly, the 50 mg/L chromium solutions were partitioned into 25 mL portions and were pH-adjusted to levels of about 3, after which they were transferred into centrifuge tubes containing peat at different doses. The peat doses of the solutions were adjusted to 0.01–1.00 g. The samples were shaken at the rate of 300 rpm at room temperature of 25 ± 1 °C. The impact of temperature was tested and verified by setting the temperatures to 10, 20, 30, 40, 50, and 60 °C. The 50 mg/L chromium solutions were partitioned into 25 mL portions with 0.08 g of modified peat in each tube and were pH-adjusted to levels of about 3 and were shaken at the rate of 300 rpm.

To assess the influence of the factors on the adsorption of the samples, an L$_{16}$(4$^4$) orthogonal experiment with four factors and four levels was designed [39], which is shown in Table 1. The 16 samples in the conditions of the experimental design were oscillated for 4 h at a rate of 300 rpm before they were centrifuged. Their chromium ion concentrations were then measured at equilibrium. The data were analyzed with SPSS statistics (IBM 21, Microsoft, Redmond, WA, USA). The degree of influence of various factors can be obtained through the orthogonal experiment and variance analysis.

**Table 1.** Factors and levels of L16(44) experiments.

| Level | Initial Concentration (mg/L) | Temperature (°C) | pH | Peat Dose (g) |
|---|---|---|---|---|
| 1 | 50 | 25 | 3 | 0.03 |
| 2 | 100 | 35 | 5 | 0.05 |
| 3 | 150 | 45 | 7 | 0.07 |
| 4 | 200 | 55 | 9 | 0.09 |

## 3. Results and Discussion

*3.1. The Selection of the Sulfonated Peat*

The present study produced three kinds of peat using three modification methods, and their different Cr(VI) removal efficiency was determined through adsorption experiments (Figure 1). Of these modified peats, all of which showed a higher adsorption efficiency than the natural peat, modified peat 1 exhibited the best results in the different Cr(VI) stock solutions. This means that oxygen, nitrogen, and sulfur functional groups were generated, creating active sites on the peat, thus strengthening its adsorption capacity [33]. The only difference among the modified peat was the temperature of the sulfonation. The experiment results indicated that higher modification temperature resulted in lower adsorption efficiency. A higher modification temperature may destroy the peat's porosity structure and impede access of the adsorbate to the functional groups. In previous studies, researchers analyzed the functional group desorption in sulfonated humic acid at different temperatures through thermogravimetric analysis (TGA) [27]. According to their research, the peak at 170 °C presents the removal process of the lignite sulfonated humic acid functional group (hydroxyl, carboxyl, phenol hydroxyl group, etc.) [40,41]. In this study, peat sulfonated at a high temperature may trigger functional group desorption. Given all of the above, modified peat 1 was selected as the object of the rest of the study. In the following research and analysis, the modified peat refers to modified peat 1.

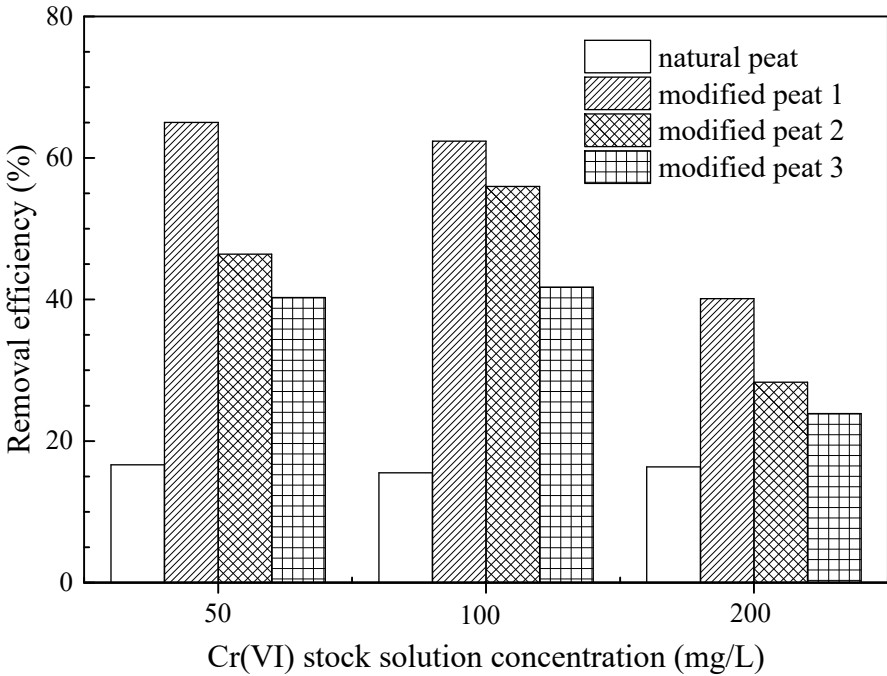

**Figure 1.** Chromium removal efficiency of modified peat using different methods. (The conditions were as follows: pH of 6, peat dose of 10 g/L, room temperature.).

*3.2. Characterization of Peat Samples*

As an adsorbent for heavy metal ions, modified peat has surface morphology and particle structure that can be directly observed by SEM and they have an immediate bearing on its adsorption properties (Figure 2). The morphologies of the natural, modified, and Cr-loaded modified peat were successively magnified by 500, 1000, and 1500 times to characterize the differences at different levels of magnification. Under 500 times magnification, the SEM images of the modified peat exhibited a polyhedral aggregate structure that was similar but smaller than the aggregate structure of its natural counterpart, and it also showed a longer distance between its aggregates. Under 1000 times magnification, the SEM images indicated that the natural peat had very little porosity and its structures were tubulous in shape. The pores of the modified peat seemed to be present inside larger macropores, but these appeared to be filled with smaller structures, which may explain the observed decrease in pore volume. The SEM images under 1500 times magnification showed that the pore size of the modified peat was bigger than that of the natural peat, which may be a result of sulfuric acid treatment as it generally enlarges the pore size of material surfaces. However, the SEM images did not indicate any remarkable differences between the modified peat and the Cr-loaded modified peat. In general, a change in the surface area of an adsorbent changes its adsorption efficiency. That is, larger specific surface areas have stronger adsorption capabilities. The surface area, pore volume, and pore size differences between the natural and modified peats are shown in Table 2. The surface area and pore volume of the modified peat were smaller than the natural peat, indicating that the physical adsorption ability of the former should have decreased. However, the modified peat exhibited a greater adsorption efficiency to Cr(VI) than the natural peat. Therefore, based on the above results, it is suggested that chemisorption dominated the adsorption process [1,40,42].

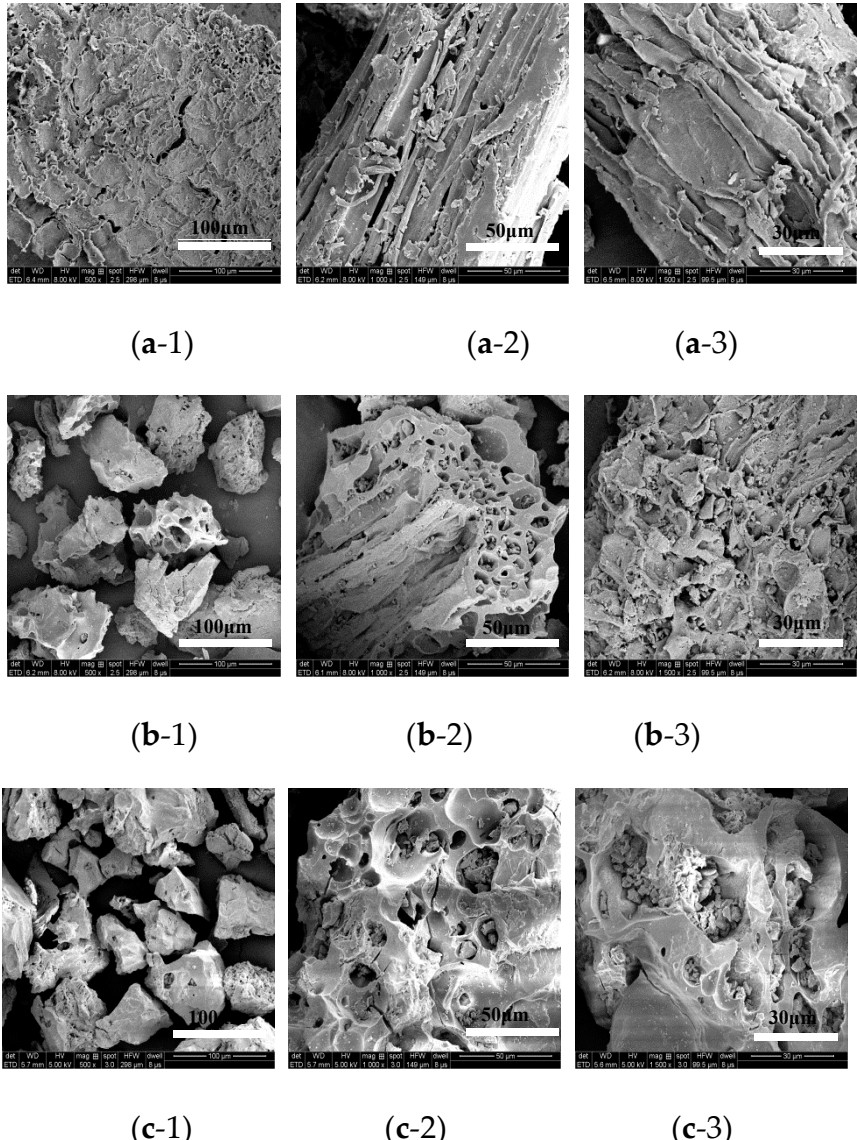

**Figure 2.** Scanning electron microscopy images of the (**a**) natural peat, (**b**) modified peat, and (**c**) Cr-loaded peat at (1) 500 x, (2) 1000 x, and (3) 1500 x magnification.

**Table 2.** Specific surface areas, pore volumes and pore sizes of the natural versus the modified.

| Peat Type | Surface Area (m$^2$/g) | Pore Volume (cm$^3$/g) | Pore Size (nm) |
|---|---|---|---|
| Natural peat | 2.49 | 0.0055 | 10.73 |
| Modified peat | 1.82 | 0.0041 | 16.22 |

The FT-IR spectra of the natural, modified, and Cr-loaded modified peat are presented in Figure 3, wherein the band intensity of the modified peat was stronger than that of the natural one. The band position at 3412 cm$^{-1}$ represented O-H stretching vibration, possibly due to water adsorption during FT-IR determination. CH$_2$ asymmetric stretching vibration was observed at 2915 cm$^{-1}$ and 2842 cm$^{-1}$, wherein the alkanes are the functional group. The peak at 1615 cm$^{-1}$ defines the C=C stretching vibration in the olefins or aromatic nuclei. The peak at 1377 cm$^{-1}$ is assumed to define the C-H bending vibration. The peak at 795 cm$^{-1}$ defines the C-H bending vibration in the aldehyde group, the peak at 630 cm$^{-1}$ defines the C-H bending vibration in-plane, and the peak at 564 cm$^{-1}$ defines the O-H bending vibration out-of-plane in the hydroxyl groups. The peak position is basically the same, the position of some peaks is offset, but the spectral band strength of sulfonated peat is stronger than

that of natural peat. At a band intensity of 3412 cm$^{-1}$, 2842 cm$^{-1}$, and 1615 cm$^{-1}$, the natural peat was weaker than that of the modified one. New vibration peaks appeared in modified peat at a band intensity of 795 cm$^{-1}$, 630 cm$^{-1}$ and 564 cm$^{-1}$. The band position at 1169 cm$^{-1}$ presents the sulfonic acid group vibration [43], wherein the band value of the modified peat was greater than the natural peat, thereby validating the successful production of sulfonated peat. The FT-IR analysis results suggest the existence of an aromatic structure as well as abundant oxygen groups such as carboxyl, carbonyl, an ester group, and an aldehyde group, among others. Peat is a natural macromolecular mixture consisting of one or more dense ring aromatic nuclei randomly connected by a plurality of functional groups through bridges and bonds [33]. These active functional groups indicate that modified peat is a more preferable adsorbent compared to unmodified peat.

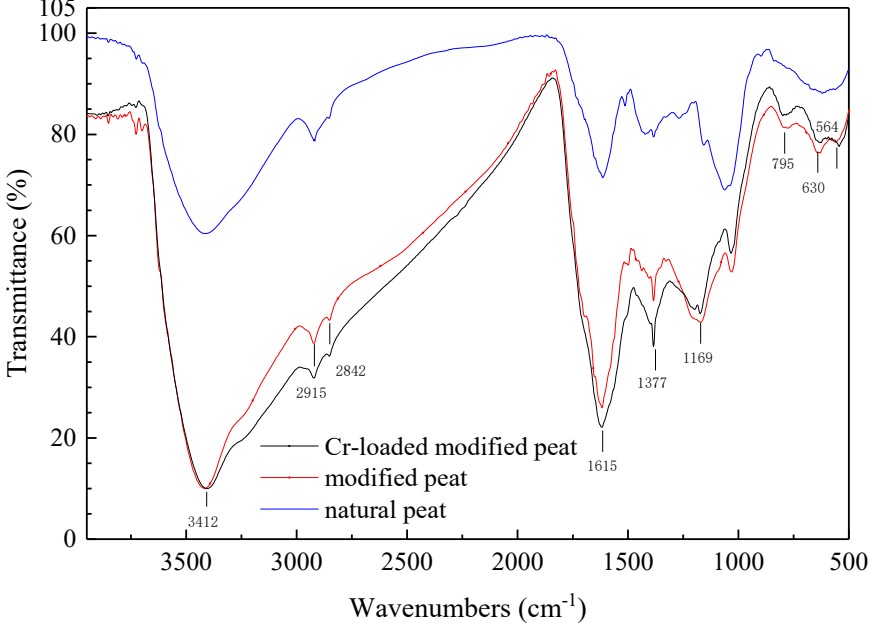

**Figure 3.** FT-IR spectra of the natural, modified, and Cr-loaded modified peat.

### 3.3. Study of the Adsorption Equilibrium of Cr(VI)

The adsorption isotherms of the modified peat for Cr(VI) removal in solution are shown in Figure 4. In general, the adsorption isotherm describes the equilibrium between the adsorbate and adsorbent in a specific set of experimental conditions. Therefore, the actual conditions for applying the adsorbent can be optimized using the adsorption isotherm parameters.

Many adsorption models are currently at our disposal, of which two isothermal adsorption models were used in this study, the Langmuir and Freundlich adsorption models [44].

The Langmuir adsorption model assumes that the single layer surface of the adsorbent dominates uptake with homogeneous adsorption [36].

$$q_{e} = \frac{K_{L}q_{max}C_{e}}{1 + K_{L}C_{e}} \tag{2}$$

where $C_e$ is the concentration of the adsorbate in the solution at equilibrium (mg/L), $q_e$ is the adsorbed amount at equilibrium (mg/g), $q_{max}$ is the saturated adsorption capacity of the adsorbent at the Langmuir monolayer adsorption state (mg/g), and $K_L$ is the constant of the Langmuir equation (L/mg).

The Freundlich adsorption model is an empirical model used to depict single- or multi-layered adsorption processes in heterogeneous systems. According to the calculation of each parameter, the

adsorption of different adsorption systems and reversible processes can be described, and is not limited by single-layer adsorption [29].

$$q_e = K_F C_e^{1/n} \tag{3}$$

where $q_e$ (mg/g) and $C_e$ (mg/L) represent the concentration of the adsorbate on the adsorbent and in the solution, respectively, and $K_F$ ($L^{1/n}$ $mg^{(1-1/n)}$/g) and n are constants that define the adsorption capacity and the energy of the adsorbent, respectively, and their values are determined by the types and properties of the given adsorbent and adsorbate [45].

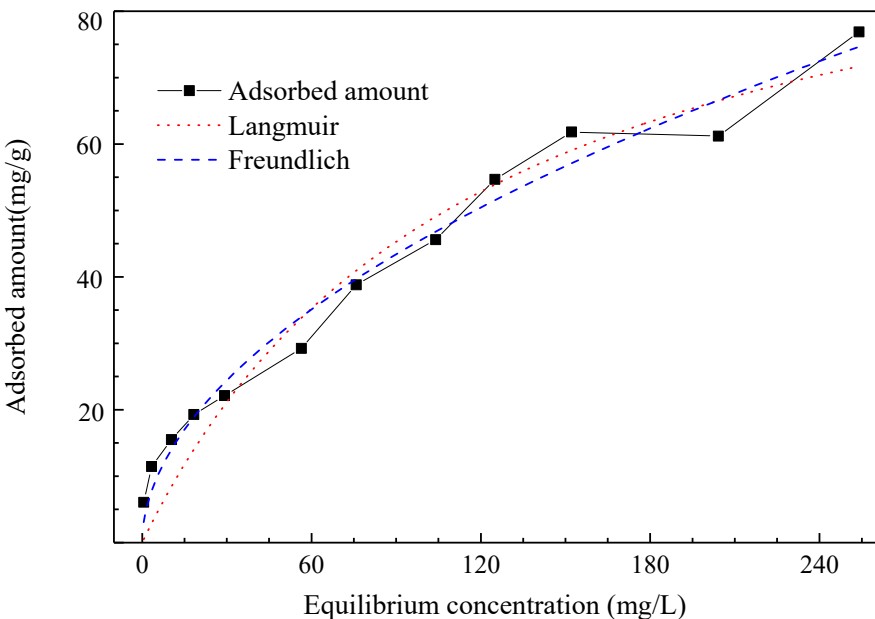

**Figure 4.** Adsorption isotherms of the modified peat with the Langmuir (red) and Freundlich (blue) models.

The Langmuir and Freundlich adsorption isotherm models were fitted to the experimental data, and the results are presented in Figure 4 and Table 3. Good fitting can be achieved using both models although the Freundlich model performs better. The adsorption of the modified peat is not limited to monolayer adsorption since with the increase in the adsorbed amount and the decrease in available adsorption sites at the surface, the adsorbate is adsorbed in new layers [5]. In the Langmuir isothermal adsorption model, the maximum adsorption capacity is calculated to reach 105.4 mg/g. According to Kołoczek et al. [32], the adsorption capacity of Cr(VI) in Canadian peat (reported as 18.75 mg/g) was significantly inferior to the values presented in this study. Research through a column experiment using sphagnum moss peat achieved a Cr(VI) adsorption capacity of 65.87 mg/g, which currently cannot be achieved with natural peat [33]. Therefore, compared to the adsorption properties of peat in other studies, the adsorption properties of sulfonated peat are significantly better.

**Table 3.** Isotherm parameters.

| Isotherm Model | Modified Peat |
|---|---|
| Langmuir | |
| $R^2$ | 0.95 |
| $K_L$ (L/mg) | 0.0084 |
| $q_{max}$ (mg/g) | 105.4 |
| Freundlich | |
| $R^2$ | 0.98 |
| $K_F$ ($L^{1/n}$ $mg^{(1-1/n)}$/g) | 4128.5 |
| $n$ | 1.91 |

### 3.4. Study of the Adsorption Kinetics

The adsorbed amount versus time graph is presented in correspondence with the uptake kinetics graph (Figure 5). The charts suggest that the adsorption proceeds for 4 h until the adsorption equilibrium is reached. The linear forms of the pseudo-first adsorption kinetic, pseudo-second adsorption kinetic, and Elovich models of $Cr^{6+}$ adsorption by the modified peat are presented in Figure 5. The data displayed in Table 4 proved that the pseudo-second order model fits the experimental results well [24]. The fitting coefficients of the pseudo-second order kinetic model exceeded 0.99, which is superior to other models.

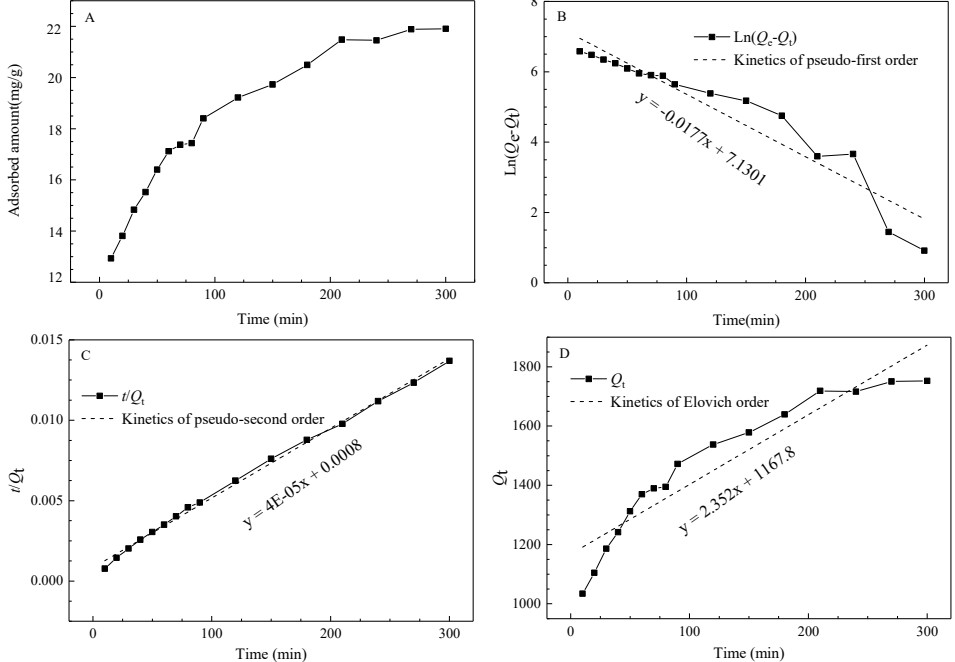

**Figure 5.** (**A**) Adsorption kinetics, with (**B**) pseudo-first order, (**C**) pseudo-second order and (**D**) Elovich order kinetic plot of Cr adsorption in the aqueous solution.

**Table 4.** Result of the orthogonal experiment on the removal efficiency influencing factors.

| Number | Initial Concentration (mg/L) | Temperature (°C) | pH | Peat Dose (g) | Removal Efficiency (%) |
|---|---|---|---|---|---|
| 1 | 50 | 25 | 3 | 0.03 | 48.24 |
| 2 | 50 | 35 | 5 | 0.05 | 26.18 |
| 3 | 50 | 45 | 7 | 0.07 | 15.9 |
| 4 | 50 | 55 | 9 | 0.09 | 13.95 |
| 5 | 100 | 25 | 5 | 0.07 | 27.08 |
| 6 | 100 | 35 | 3 | 0.09 | 52.32 |
| 7 | 100 | 45 | 9 | 0.03 | 5.99 |
| 8 | 100 | 55 | 7 | 0.05 | 16.2 |
| 9 | 150 | 25 | 7 | 0.09 | 19 |
| 10 | 150 | 35 | 9 | 0.07 | 11.66 |
| 11 | 150 | 45 | 3 | 0.05 | 30.57 |
| 12 | 150 | 55 | 5 | 0.03 | 25.08 |
| 13 | 200 | 25 | 9 | 0.05 | 15.64 |
| 14 | 200 | 35 | 7 | 0.03 | 16.45 |
| 15 | 200 | 45 | 5 | 0.09 | 35.63 |
| 16 | 200 | 55 | 3 | 0.07 | 46.67 |
| K1 | 26.07 | 27.49 | 44.45 | 23.94 | |
| K2 | 25.4 | 26.65 | 28.5 | 22.15 | |
| K3 | 21.58 | 22.02 | 16.89 | 25.33 | |
| K4 | 28.6 | 25.48 | 11.81 | 30.22 | |
| R | 7.02 | 5.47 | 32.64 | 8.07 | |

The pseudo-first order kinetic plot follows Equation (4),

$$\frac{dQ_t}{dt} = k_1(Q_e - Q_t) \tag{4}$$

which can be transformed into Equation 5.

$$\ln(Q_e - Q_t) = lnQ_e - k_1 t \tag{5}$$

The pseudo-second order kinetic plot follows Equation 6,

$$\frac{dQ_t}{dt} = k_2(Q_e - Q_t)^2 \tag{6}$$

which can be transformed into Equation 7 [35].

$$\frac{t}{Q_t} = \frac{1}{k_2 Q_e^2} + \frac{t}{Q_e} \tag{7}$$

In these equations, $Qe$ is the adsorbed amount at equilibrium (mg/g), $k_1$ is a given constant reflecting the adsorption rate of pseudo-first order kinetic model ($min^{-1}$), $k_2$ is a given constant reflecting the adsorption rate of the pseudo-second order kinetic model ($min^{-1}$), $t$ is the adsorption time (min), and $Q_t$ is the adsorbed amount at a specific time t (mg/g).

The Elovich model follows Equation 8,

$$q_t = \frac{1}{\beta}\ln(\alpha\beta) + \frac{1}{\beta}\ln t \tag{8}$$

In this equation, $\alpha$ and $\beta$ are the coefficients of the equation, $\alpha$ is the initial adsorption rate (mg/ (g·h)), and $\beta$ is the reciprocal of the surface coverage when the adsorption rate is 1/e of its initial value (g/mg) [46], $q_t$ is the adsorbed amount at a specific time t (mg/g), and $q_e$ is the adsorbed amount at equilibrium (mg/g). The linear forms of these three models of $Cr^{6+}$ adsorption by the modified peat and the values of $k$ and $R^2$ (correlation coefficient) calculated from the experimental results are presented in Table 4.

According to Figure 5A, the adsorbed amount at equilibrium is approximately 22 mg/g. The pseudo-second order model is established in the entire time range of the adsorption equilibration, which involves the diffusion of the liquid membrane on the peat surface, surface diffusion, and adsorption. The model assumes that adsorption follows a second order mechanism and that the adsorption is dominated by chemisorption. This chemical reaction involves electron sharing between the functional groups of the adsorbent and heavy metal ions, or chemical adsorption in the form of electron gain or loss [44].

### 3.5. Influences of pH, Adsorbent Dose, and Temperature

The adsorbent dose is an important and influential factor in the removal efficiency. In general, the more modified peat is added, the more Cr is removed from the solution. The adsorption of chromium increases with an increase in the peat dose, possibly due to the greater surface area and abundance of adsorption sites on the sorbent [36]. However, given the economic factors, the smallest yet optimal amount of peat dose must be determined. Hence, the influence of the peat dose on the removal efficiency was evaluated and the results are displayed in Figure 6A. The removal efficiency kept increasing as the adsorbent dose increased until an adsorbent dose of 8 g/L. Therefore, the optimal amount of added modified peat was identified as 0.2 g per 25 mL of Cr(VI) solution.

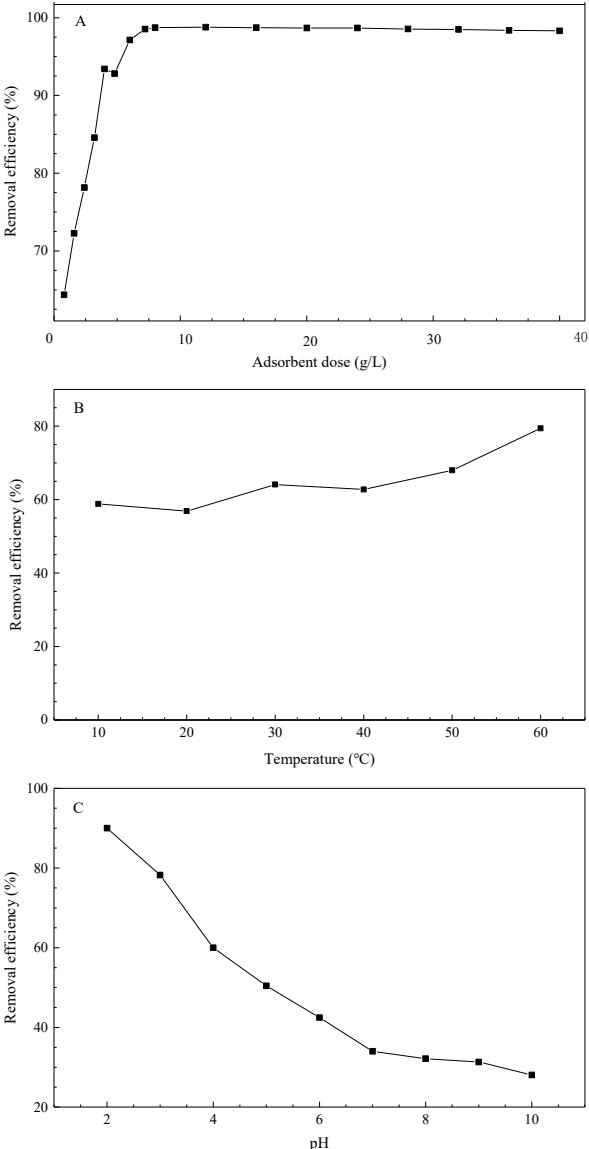

**Figure 6.** Effects of the (**A**) adsorbent dose, (**B**) temperature, and (**C**) pH on the removal of Cr(VI) by the modified peat.

The temperature exhibited beneficial effects on the adsorption process. Figure 6B implies that the adsorption increased slightly with the rise in temperature, thereby suggesting it is an endothermic process. Therefore, in practical applications, if conditions allow, waste heat can be utilized to raise the adsorption temperature for enhanced adsorption.

The pH significantly influences the removal of heavy metals as it can change the adsorbent's surface charges aside from altering the solution's chemical properties. Figure 6C shows the effects of pH, that is, as the pH value increases, the removal of Cr(VI) decreases. In addition, at pH values exceeding 7, the decrease stabilizes [33]. The zeta potentials of the modified peat at different pH levels were measured and are presented in Figure 7, and they show that the modified peat is negatively charged when its pH falls within the range of 2 to 9. With the increase in pH, the negative charge of sulfonated peat increased, and the electrostatic adsorption of sulfonated peat to dichromate was decreased. Meanwhile, the increase in Zeta potential increased the degree of agglomeration and decreased the dispersibility of peat surface particles, which reduces the probability of collision between adsorbent particles and Cr (VI) ions.

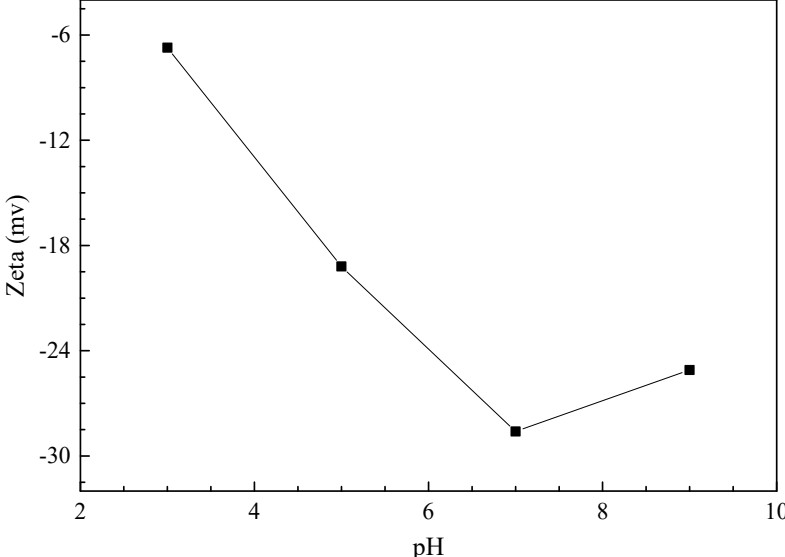

**Figure 7.** Zeta potentials of the modified peat.

The adsorbed $Cr^{6+}$, part of which remains firmly in the peat, and part of which is reduced to $Cr^{3+}$ by the peat, as $Cr^{6+}$ has strong oxidative power under acidic conditions; it can react with reductive groups present in humic substances in the modified peat such as the phenolic hydroxyl group, alcohol hydroxy, and so on. This process can be expressed using the equation below, wherein C represents various functional groups in the organic matter in this experiment. The $Cr^{6+}$ anions contained in the solution can be reduced to $Cr^{3+}$ when diffused through the pores on the modified peat. [15].

$$3C + 2Cr_2O_7{}^{2-} + 16H^+ \rightarrow 4Cr^{3+} + 3CO_2 + 8H_2O \tag{9}$$

The above three factors exerted different influences on the removal efficiency. This study also delved into the effects of adsorption under different environmental conditions through a four-level orthogonal test on the four factors. $K$ represents the mean value of a certain level of a factor, $K_1$ represents the average removal efficiency at first level of each influencing factor, and so on. $R$ represents the range value of $K$. The larger the $R$ value is, the greater the influence of this factor on the test results. By comparing the magnitude of the $R$ value, the factors can be sorted by the magnitude of their influence on the results.

By means of variance analysis, with a significance level $\alpha = 0.05$, $F\alpha$ (3, 3) = 9.28. The analysis showed that the $F$(initial concentration) > 9.28, and this indicates that the initial concentration had a significant impact on the removal efficiency of Cr from sulfonated peat, while the $F$(temperature) < 9.28, suggesting that temperature had no significant impact on the removal efficiency of Cr by sulfonated peat. Moreover, $F$(pH) > 9.28, and this indicates that pH had a significant impact on the removal efficiency of Cr by sulfonated peat, and $F$(peat dose) > 9.28, indicating that the peat dose also had a significant impact on the removal efficiency of Cr from sulfonated peat.

According to the results of the orthogonal test and the range analysis (Table 5), the various influencing factors were sorted based on their influence: pH > peat dose > initial concentration > temperature. pH is the most significant factor that affects the adsorption removal efficiency whereas the initial concentration, temperature and peat dose generated almost equal effects on efficiency.

**Table 5.** Variance analysis of orthogonal experimental results.

| Source of Variation | Sum of Squares of Type III | Degrees of Freedom | F | Significance |
|---|---|---|---|---|
| Initial concentration | $6.7 \times 10^8$ | 3 | 27.840 | 0.011 |
| Temperature | $9.3 \times 10^7$ | 3 | 3.784 | 0.152 |
| pH | $5.4 \times 10^9$ | 3 | 22.244 | 0.015 |
| Peat dose | $2.3 \times 10^8$ | 3 | 9.451 | 0.049 |
| Error | $2.4 \times 10^7$ | 3 | | |

## 4. Conclusions

The adsorption efficiency of the modified peat was higher than that of its natural counterpart, and the best modification method required the sulfonation of peat for 24 h at room temperature while maintaining a solid-to-liquid ratio of 1:4.

The modified peat is a strong adsorbent for Cr(VI), and a maximum adsorption capacity of 105.4 mg/g was obtained. The pseudo-second-order model fit the adsorption kinetics. The results indicated that the adsorption of chromium by modified peat is a multi-stage process. According to the conducted adsorption experiments, the adsorption process followed the Freundlich isotherm adsorption model and the adsorption of chromium by the modified peat was not limited to single layer adsorption.

In addition, according to the variance analysis, the effect of temperature is not significant, and the effect of other factors is significant, among which pH has the largest effect.

**Author Contributions:** The idea for the study. was provided by H.C. and Y.C. The design of the experiment method, the control of workflow, formal analysis, investigation and the writing of the paper were accomplished by H.L. and R.H., R.H. conducted the pretreatment of samples. H.L. carried out instrument analysis. H.C. and Y.C. administrated the project and reviewed manuscript.

**Funding:** This research was funded by the National Science and Technology Major Project of China, grant number 2017ZX07301005-003. This research was also supported by the National Science and Technology Major Project of China, grant number 2017ZX07107-005.

**Conflicts of Interest:** The authors declare no conflict of interest.

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
