# Peer review of "Removal of Hexavalent Chromium from Aqueous Solutions Using Sulfonated Peat"

_water, doi:10.3390/w11101980_

Round 1

Reviewer 1 Report

The paper has been greatly improved but there is one methodological inconsistency that remains. On line 314, the best adsorbent dosage is indicated as 8 g/L, corresponding to an absolute dosage of 0.2 g of peat, however in the orthogonal experiments the peat dose is varied between 0.03 and 0.09 g. Similarly, the process seems to be most efficient at 60ºC temperature and pH of 2 but the temperature is varied between 25-55ºC and the pH between 3-9. Finally, including the “initial concentration” as a factor in the removal efficiency is, in my opinion, not a good option since this is a variable of which there is little control in a real treatment setting. If we are to optimize treatment, we should do so for the variables which we can control and that we can apply to most conditions of an influent.

In light of these observations, I absolutely cannot agree with the authors’ conclusions presented on lines 20-21, 358-359, and 375-376. The rest of the study contradicts these as the optimal conditions for treatment. The authors must either a) redo the orthogonal experiments in light of the conclusions drawn from single factor experiments, that is, selecting factors and ranges that are consistent with the rest of the study; b) present the current orthogonal experiment without any ‘optimized’ results, but only as an indication of the degree of influence of each factor (though I have doubts about the relevance of such results); or c) not present the results of the orthogonal experiments altogether, deriving the conclusions solely from single-factor experiments, so that the manuscript is consistent from beginning to finish.

Some other observations, corrections and/or suggestions:

In the interpretation of FTIR spectra, the band of 1169 cm-1 is indicated as both C-O-C asymmetric stretching vibration and sulfonic acid group vibration. Having in consideration the near inexistence of this band on natural peat, I would interpret this band as being solely the sulfonic acid group vibration. Similarly, the band at 1035 cm-1 does not seem to correspond to sulfonic group, since it is greater on natural than sulfonated peat (we must take into consideration the shape of the peak, and the offset of the spectra) – therefore the conclusions given on lines 228-229 are only valid for 1169 cm-1 band and not 1035 cm-1. 4 and 5 would become more readable if the experimental points were not united by lines. Lines 265-266: instead of “with the adsorption capacity increase and the decrease in available adsorption sites at the surface, adsorbate enter the next layers”, write “since with the increase of adsorbed amount and the decrease in available adsorption sites at the surface, the adsorbate is adsorbed in new layers” Lines 290-292: please uniformize units throughout the paper, if you are presenting some as mg/g, please use mg/(g.h) and g/mg as the units for α and β. Line 294: correct terminology of r2 is ‘correlation coefficient’ Line 299: instead of “adsorption capacity” use “adsorbed amount at equilibrium” Line 302: instead of “the sorption” write “adsorption” Line 305: not sure what is the point of this sentence: “In addition, the adsorption process are chemical adsorption processes [44].”, I suggest removing it Line 337: instead of “As Cr6+ has strong oxidation under acidic conditions, it can react with such reductive groups”, write “as Cr6+ has strong oxidative power under acidic conditions, it can react with reductive groups” Lines 347-348: instead of “the ability to influence factors can be sorted”, write “the factors can be sorted by the magnitude of their influence on the results”

I hope you can find my comments of interest for the improvement of the paper.

Author Response

Response to Reviewer 1 Comments

Point 1: The paper has been greatly improved but there is one methodological inconsistency that remains. On line 314, the best adsorbent dosage is indicated as 8 g/L, corresponding to an absolute dosage of 0.2 g of peat, however in the orthogonal experiments the peat dose is varied between 0.03 and 0.09 g. Similarly, the process seems to be most efficient at 60ºC temperature and pH of 2 but the temperature is varied between 25-55ºC and the pH between 3-9. Finally, including the “initial concentration” as a factor in the removal efficiency is, in my opinion, not a good option since this is a variable of which there is little control in a real treatment setting. If we are to optimize treatment, we should do so for the variables which we can control and that we can apply to most conditions of an influent.

In light of these observations, I absolutely cannot agree with the authors’ conclusions presented on lines 20-21, 358-359, and 375-376. The rest of the study contradicts these as the optimal conditions for treatment. The authors must either a) redo the orthogonal experiments in light of the conclusions drawn from single factor experiments, that is, selecting factors and ranges that are consistent with the rest of the study; b) present the current orthogonal experiment without any ‘optimized’ results, but only as an indication of the degree of influence of each factor (though I have doubts about the relevance of such results); or c) not present the results of the orthogonal experiments altogether, deriving the conclusions solely from single-factor experiments, so that the manuscript is consistent from beginning to finish.

Response 1: Thanks for your comments. As you said, the range of influencing factors in the orthogonal experiment is not perfect. We agree with the plan b you said, present the current orthogonal experiment without any optimized results, but only as an indication of the degree of influence of each factor. Because we think this experiment has certain significance in the future study we can give priority to control the factors that have great influence. So we don't want to give that up. Thank you again for your kind comments.

Point 2: In the interpretation of FTIR spectra, the band of 1169 cm-1 is indicated as both C-O-C asymmetric stretching vibration and sulfonic acid group vibration. Having in consideration the near inexistence of this band on natural peat, I would interpret this band as being solely the sulfonic acid group vibration. Similarly, the band at 1035 cm-1 does not seem to correspond to sulfonic group, since it is greater on natural than sulfonated peat (we must take into consideration the shape of the peak, and the offset of the spectra) – therefore the conclusions given on lines 228-229 are only valid for 1169 cm-1 band and not 1035 cm-1.

Response 2: Thanks for your comments. This analysis has been changed in the revised manuscript.

Point 3: 4 and 5 would become more readable if the experimental points were not united by lines.

Response 3: Thanks for your comments. By convention, a lot of the literature is drawn this way. But we made some color changes.

Point 4: Lines 265-266: instead of “with the adsorption capacity increase and the decrease in available adsorption sites at the surface, adsorbate enter the next layers”, write “since with the increase of adsorbed amount and the decrease in available adsorption sites at the surface, the adsorbate is adsorbed in new layers”

Response 4: Thanks for your comments. This sentence has been changed in the revised manuscript.

Point 5: Lines 290-292: please uniformize units throughout the paper, if you are presenting some as mg/g, please use mg/(g.h) and g/mg as the units for α and β.

Response 5: Thanks for your comments. These units has been changed in the revised manuscript.

Point 6: Line 294: correct terminology of r2 is ‘correlation coefficient’

Response 6: Thanks for your comments. This terminology has been corrected in the revised manuscript.

Point 7: Line 299: instead of “adsorption capacity” use “adsorbed amount at equilibrium”

Response 7: Thanks for your comments. "adsorption capacity" has been changed to "adsorbed amount at equilibrium" in the revised manuscript.

Point 8: Line 302: instead of “the sorption” write “adsorption”

Response 8: Thanks for your comments. "the sorption" has been changed to "adsorption" in the revised manuscript.

Point 9: Line 305: not sure what is the point of this sentence: “In addition, the adsorption process are chemical adsorption processes [44].”, I suggest removing it

Response 9: Thanks for your comments. This sentence has been deleted in the revised manuscript.

Point 10: Line 337: instead of “As Cr6+ has strong oxidation under acidic conditions, it can react with such reductive groups”, write “as Cr6+ has strong oxidative power under acidic conditions, it can react with reductive groups”

Response 10: Thanks for your comments. This sentence has been changed in the revised manuscript.

Point 11: Lines 347-348: instead of “the ability to influence factors can be sorted”, write “the factors can be sorted by the magnitude of their influence on the results”

Response 11: Thanks for your comments. This sentence has been changed in the revised manuscript.

Thanks again for your great comments. Having gone through so many revisions, your suggestions are crucial to my article. We will fully consider many of your suggestions in future research.

Reviewer 2 Report

This is an extremely reasonable describing a rigorous and incremental advance in the use of peat for the adoption of heavy metals.

I don’t know the journals policy, but I would suggest moving the figure label in front of what it describes in Figure 2, 5, and 6. For example: “Figure 5. (A) Adsorption kinetics” rather than “Figure 5. Adsorption kinetics (A)” This would be much easier to read. There are no colors in Figure 4. Way too much is included concluded from the FT-IR data in Figure 3. I’m not confident the same thickness of sample was used for all three samples. They may all like essentially the same if the same thickness was used. Please eliminate the discussion that precedes Figure 3. Figure 2. I’m not confident that Figure a-2 and 2-3 is looking at the same material cross section observed in b-2 and b-3 as well as c-2 and c-3.

In summary this is an extremely reasonable paper if not particular exciting. The author draws more conclusions from the SEM and IR data than is strictly reasonable but the paper’s results in no way hinges on the interpretation of these results. The adsorption data is interpenetrated soundly.

Author Response

Response to Reviewer 2 Comments

Point 1: I don’t know the journals policy, but I would suggest moving the figure label in front of what it describes in Figure 2, 5, and 6. For example: “Figure 5. (A) Adsorption kinetics” rather than “Figure 5. Adsorption kinetics (A)” This would be much easier to read.

Response 1: Thanks for your comments. This format has been changed in the revised manuscript.

Point 2: There are no colors in Figure 4.

Response 2: Thanks for your comments. We have added colors in Figure 4 in the revised manuscript.

Point 3: Way too much is included concluded from the FT-IR data in Figure 3. I’m not confident the same thickness of sample was used for all three samples. They may all like essentially the same if the same thickness was used. Please eliminate the discussion that precedes Figure 3. 

Response 3: Thanks for your comments. Yes, these three samples are essentially the same, all three samples are in powder form, and treated in same conditions, but because of the sulfonation modification, the functional group changes, the FT-IR test functional group changes. 

Point 4: Figure 2. I’m not confident that Figure a-2 and 2-3 is looking at the same material cross section observed in b-2 and b-3 as well as c-2 and c-3.

Response 4: Thanks for your comments. We did SEM of three samples at the same site with different magnification.

Thanks again for your great comments.

Round 2

Reviewer 1 Report

-

This manuscript is a resubmission of an earlier submission. The following is a list of the peer review reports and author responses from that submission.

Round 1

Reviewer 1 Report

This is my second time reviewing this manuscript. The authors made some significant improvements on the previous version, however, many issues remain that need to be addressed by the authors before the article becomes suitable for publication.

The English expression was improved in the instances were errors were pointed out; however, a thorough revision of the manuscript is recommended, since there are parts where the inaccuracy of the grammar compromised the clarity of the points that you were trying to get across.

Some inconsistencies with the terminology, presentation and discussion of results also need to be clarified and/or corrected.

Abstract

·         Lines 10-11: instead of “can be used as adsorbents” write “can be used as an adsorbent”

·         Line 16: please review the significant digits in this number (105.404 mg/g) since it is highly unlikely that the value is precise to three decimal digits. If possible, include the uncertainty of the value.

·         Lines 19-21: you present here contradictory information, since you refer as optimal dose of peat both 3200 mg/L and 3.6 g/L. I understand that the latter is provided by the orthogonal design, but you cannot present both results as the best dosage since it is inaccurate.

Introduction

·         Line 27: instead of “the least neglected” write “one of the most important”

·         Line 28: instead of “industrial activities such as electroplating” write “industrial activities, from electroplating”

·         Line 31: instead of “with the help of rainwater of which Cr(III) and Cr(VI)” write “with the help of rain water. Cr(III) and Cr(VI)”

·         Line 36: instead of “increased attentions have been put on its damages” write “increased attention has been drawn to its damages”

·         Line 38: instead of “reported the capability” write “reported on the capability”

·         Line 41: instead of “many researches” write “many studies”

·         Line 43: instead of “due to high cost” write “due to their high cost”

·         Line 43: instead of “recent researches” write “recent studies”

·         Line 48: instead of “contains large quantity” write “contains a large quantity”

·         Lines 48-49: instead of “humic substances, and has a large” write “humic substances, has a large”

·         Line 53: instead of “are traced back” write “can be traced back”

·         Line 54: instead of “researches” write “studies”

·         Line 57: instead of “have been examined” write “have been examining”

·         Line 60: instead of “as the adsorbent” write “as adsorbent”

·         Line 64: instead of “when used” write “when it is used”

·         Line 68: instead of “following 4.9 ± 0.01 mg/g of peat” write “followed by 4.90 ± 0.01 mg/g in peat”

·         Line 69: instead of “18.6 mg/g of peat” write “18.6 mg/g in peat”

·         Line 73: instead of “increased attentions are focused” write “further research is being focused”

·         Lines 74-75: This has been mentioned in the previous review, this sentence is contradictory with what was presented earlier in the paragraph. I would review or remove it.

·         Line 80: instead of “kinetics characteristics were evaluated. The interaction mechanism of Cr(VI) to” write “kinetics and the interaction mechanism of Cr(VI) with”

·         Line 81: instead of “was also examined” write “were also examined”

Materials and Methods

·         Line 94: instead of “were then tested, whereupon the best is identified”, write “was then tested, in order to identify the best performing one”.

·         Line 111: instead of “using FT-IR and by analyzing changes” write “using FT-IR by analyzing changes”

·         Line 114: please review the significant digits according to the uncertainty. If the uncertainty has three decimal digits then the value should have as well, e.g. 0.030 ± 0.002 g

·         Lines 114-117: refer the pH at which these experiments were carried out

·         Line 118: instead of “adsorbing capacity”, write “adsorbed amount”. Also, the terminology for adsorbed amount is usually “q” and not “Y”.

·         Line 118-119: eliminate “of which the experimental data was fitted using the Langmuir and Freundlich isotherm models”, this is redundant, since it will be explained later in the text.

·         Line 121: instead of “after equilibrium” write “at equilibrium”

·         Line 122: instead of “chromium” write “solution”

·         Line 122: instead of “mL” write “L”

·         Line 127: same with significant digits as in line 114. Also, explain why a different dosage was used for isotherms and for kinetics?

·         Lines 127-131: refer the pH at which these experiments were carried out

·         Line 132: instead of “the adsorbances of” write “the adsorbed amounts for”

·         Lines 133-134: instead of “a pseudo-second-order kinetic model” write “pseudo-second-order kinetic model”

·         Line 140: instead of “amounts of 0.5 mol/L HNO3 and 0.5 mol/L NaOH” write “amount of 0.5 mol/L HNO3 or 0.5 mol/L NaOH”

·         Line 143: be more precise with the parameters, “about 3 and 4” is very vague.

·         Lines 140-146: explain more precisely the single factor experiments step by step for each factor, including the fixed parameters when one of the factors was varied.

·         Line 148: instead of “with 4-factor, 4-level” write “with 4 factors and 4 levels”

·         Line 148: I cannot find reference [39], and it does not support the theory of the experimental design. Please review

·         Line 148: instead of “The 16 above samples”, write “The 16 samples in the conditions of the experimental design”

Results and Discussion

·         Line 158: instead of “stock solutions thereby generationg oxygen” write “stock solutions. This means that oxygen”

·         Line 159: instead of “and creating active sites” write “were generated creating active sites”

·         Line 161: instead of “sulfonate” write “sulfonation”

·         Line 161: instead of “in a low” write “in lower”

·         Line 173: instead of “and a peat dose” write “peat dose”

·         Line 178: instead of “1500 times magnification” write “1500 times”

·         Line 181: instead of “between its aggregate” write “between its aggregates”

·         Lines 183-187: I cannot agree with the observations. The imagines do not show the fine porosity (it is nm-sized), only macropores (μm-sized), and the natural peat actually seems to show very little porosity. Similarly, the images do not have enough resolution for conclusions to be drawn about the surface smoothness. This explanation needs to be reviewed.

·         Line 191: instead of “are demonstrated”, write “are shown”

·         Line 191: eliminate “The result is beyond all expectations”, this is speculative

·         Line 193: instead of “have slackened” write “have decreased”

·         Line 204: instead of “adsorbance peat” write “Cr-loaded peat”

·         Table 2: review the significant numbers, I doubt that the numbers are significant to the fourth decimal digit or the seventh decimal digit in the case of pore volume. Also, if possible, include the uncertainties of the determinations.

·         Line 207: instead of “was basically stronger” write “was stronger”

·         Lines 210-213: In my opinion, peaks at 1699 cm-1 and 1503 cm-1 were misidentified, since they are not important features in the spectrum through the observation in the graph of Fig. 3. I’m not sure if it’s relevant for them to be mentioned in the text.

·         Line 217: On the other hand, the peak at 1035 cm-1 is not referred and is quite significant for all the acquired spectra.

·         Lines 220-221: The statement that 1035 cm-1 is a new vibration peak is inaccurate, since in Fig. 3 a peak can be seen in all spectra at this wave number, even though the shape is different. On the other hand, the peak at 1169 cm-1 is only present in the modified peat samples, and this is not referred in the text.

·         Lines 224-225: I’m not sure I agree that peaks at 1035 cm-1, 795 cm-1, and 630 cm-1 are weaker in Cr-modified peat; the only relevant difference is seen for the 1169 cm-1 peak, in my opinion.

·         Lines 230-232: How are these conclusions drawn? FTIR is not quantitative, and provides no information about the surface acidity, hydrophilicity, chelation and ion exchange properties. The only inference that could possibly be said is that due to the magnitude of the peaks, it is likely that modified peat has more functional groups available for adsorption than natural peat. Other statements are speculative.

·         Figure 3: in the legend, write “natural peat” instead of “nature peat”

·         Line 237: instead of “the interaction” write “the equilibrium”

·         Figure 4: y-axis legend should be “adsorbed amount” instead of “adsorbance”, and mg/g seems to be the more appropriate unit, given the order of magnitude of the values.

·         Line 241: instead of “in the Langmuir” write “with the Langmuir”

·         Lines 243-244: This sentence describes a normal behavior, in my opinion it is redundant to refer to this in the text

·         Lines 244-248: Very confusing statements, need thorough revision or removal from the text

·         Line 248: instead of “As a result, many adsorption models” write “Many adsorption models”

·         Line 258: instead of “adsorption thermodynamic processes of” write “adsorption processes in”

·         Lines 258-261: again very confusing statements that need thorough revision or removal from the text

·         Line 263: units of KF should be (mg g-1) (L mg-1)1/n

·         Table 3: review the significant numbers of the presented values, highly unlikely that they are relevant to the fifth decimal digit. If possible, provide uncertainty of the values of the parameters. Refer to the units of KF

·         Lines 268-269: instead of “The adsorption isotherm of the modified peat was fitted to the Langmuir and Freundlich adsorption isothermal models, of which the fitting results are presented” write “The Langmuir and Freundlich adsorption isotherm models were fitted to the experimental data, and the fitting results are presented”

·         Lines 269-270: instead of “The adsorption of Cr(VI) by the modified peat can be fitted” write “Good fitting can be achieved”

·         Line 272: instead of “the active points on the peat surface decrease” write “the decrease in available adsorption sites at the surface”

·         Lines 272-273: clarify the meaning of “adsorbate enter the granular layers” (confusing statement)

·         Line 274: instead of “to reach as high as” write “to reach”

·         Lines 274-277: the information presented for comparison is very incomplete. Which material and conditions were used in these studies?

·         Line 281: “a progressively expedited pace” – this is incorrect, actually the rate of adsorption is fast at the beginning and then slows down, please review the statement

·         Line 284: instead of “is presented” write “are presented”

·         Lines 284-285: instead of “the experimental results fit the pseudo-second-order kinetic model” write “the pseudo-second-order model fits well the experimental results”

·         Line 287: instead of “kinetic plots follows the Equation 4” write “kinetic plot follows Equation 4”

·         Line 288: instead of “deduced to” write “transformed into”

·         Line 289: instead of “kinetic plots follows the Equation 6” write “kinetic plot follows Equation 6”

·         Line 290: instead of “deduced to” write “transformed into”

·         Line 291: instead of “In these Equation, Qe is the equilibrium maximum adsorbance (mg/kg)” write “In these equations, Qe is the adsorbed amount at equilibrium (mg/g)”

·         Lines 291-292: refer to the difference between k1 (pseudo-first-order rate) and k2 (pseudo-second-order rate)

·         Lines 292-293: instead of “the adsorbance in a specific period (mg/kg)” write “the adsorbed amount at a specific time t (mg/g)”

·         Line 294: instead of “follows the Equation 8” write “follows Equation 8”

·         Line 295: describe better the meaning of α and β coefficients and refer to their units

·         Lines 295-296: instead of “the adsorbance in a specific period (mg/kg)” write “the adsorbed amount at a specific time t (mg/g)”

·         Line 296: instead of “the equilibrium adsorbance (mg/kg)” write “the adsorbed amount at equilibrium (mg/g)”

·         Line 297: eliminate “is presented”

·         Line 298: instead of “are demonstrated” write “are presented”

·         Figure 5: y-axis of graph A : instead of “adsorbance (mg/kg)” it should be “adsorbed amount (mg/g)”

·         Line 300: instead of “Adsorption kinetics (A)” write “Adsorption kinetics with (A)”

·         Table 4: there are parameters missing in this Table, it is incomplete

·         Line 306-307: “thereby revealing the adsorption mechanism” – a speculative statement, I suggest eliminating

·         Lines 308-309: instead of “the chemisorption of the valence forces or covalent forces between the adsorbent and the adsorbate” write “chemisorption”

·         Lines 309-310: instead of “the electronic sharing” write “electron sharing”

·         Line 313: instead of “adsorption process” write “adsorption kinetics”

·         Lines 312-315: very confusing statements, need revision or removal

·         Line 323: instead of “was characterized, of which the results” write “was evaluated and the results”

·         Line 324: instead of “kept accelerating” write “kept increasing”

·         Line 324: “0.08 g” – please present the dosage as concentrations instead of absolute values, for better comparison with other studies and better understanding. The same in the axis of Fig. 6.

·         Line 331: instead of “sped up in tandem with the rise” write “increased slightly with the rise”

·         Line 332: instead of “if conditions permit” write “if conditions allow”

·         Line 333: instead of “faster adsorption” write “enhanced adsorption”

·         Line 338: instead of “decelerates” write “decreases”

·         Line 339: instead of “the deceleration slows down” write “the decrease stabilizes”

·         Line 339-340: not sure what you mean by this sentence. I suggest it is removed.

·         Line 341: instead of “wherein” write “and they show that”

·         Lines 342-344: how is this conclusion drawn from the data? Also, I saw nothing in ref. 5 to support this statement. Please revise.

·         Lines 344-345: please revise this statement since surface neutralization cannot be occurring, because Cr(VI) is anionic.

·         Lines 346-348: This statement also needs revision since Cr6+ has reductive and not oxidative potential.

·         Lines 350-351: instead of “diffused to their reaction positions” write “diffused”

·         Line 353: “the rise in pH was promoted by ion exchange” – nothing in the results suggest this, please explain or revise the statement

·         Line 354: instead of “Cr6+” it should be “Cr3+

·         Lines 356-359: this statement also needs revision since it indicates that Cr removal is favoured at higher pH conditions and this is contradictory to the experimental results.

·         Table 5: the meaning of the K1, K2, K3, K4 and R parameters was not clarified.

·         Table 6: Sum of squares values are very high, please revise your results, also instead of “degree of freedom” write “degrees of freedom”

·         Line 369: instead of “the significant level” write “with a significance level”

·         Line 369: instead of “9.28, F” write “9.28. This means that since F”

·         Line 370: instead of “indicating” write “this indicates”

·         Line 371: instead of “F(temperature) < 9.28, indicating that temperature” write “while since F(temperature) < 9.28, temperature”

·         Line 372: instead of “sulfonated peat, indicating that pH” write “sulfonated peat. Moreover, F(pH) > 9.28, indicating that pH”

·         Line 373: instead of “sulfonated peat, F(addition)> 9.28” write “sulfonated peat, and F(peat dose) > 9.28”

·         Line 374: instead of “the addition of peat” write “the peat dose also”

·         Lines 375-376: the optimal conditions here presented are not consistent with the single factor experiments. You refer this for temperature on lines 376-378, but this is also valid for the other factors, and it is not explained well.

·         Line 380: instead of “their strengths”, write “their influence”

·         Lines 380-381: please explain the sorting of the influencing factors, since it does not seem to be consistent with F or significance values.

·         Lines 381-384: these statements are very confusing and need to be revised.

Conclusions

·         Line 390: “under the presented conditions” – clarify and describe these conditions

·         Lines 390-391: instead of “The kinetic equation of the adsorption process fit the pseudo-second-order kinetic model” write “The pseudo-second-order model fit the adsorption kinetics”

·         Lines 392-393: instead of “conducted isothermal adsorption experiment, the adsorption thermodynamic process” write “conducted adsorption experiments, the adsorption process”

·         Line 393: instead of “Freundlich isothermal” write “Freundlich isotherm”

·         Line 396: instead of “single factor experiment and orthogonal experiment” write “single factor experiments and orthogonal experiments”

·         Lines 396-398: these conclusions need to be revised according to the revision required in the results and discussion section.

I hope you find my comments of interest for the improvement of the manuscript.

Author Response

Response to Reviewer 1 Comments

Point 1: Lines 10-11: instead of "can be used as adsorbents" write "can be used as an adsorbent"

Response 1: Thanks for your comments. "can be used as adsorbents" has been changed to "can be used as an adsorbent" in the revised manuscript.

Point 2: Line 16: please review the significant digits in this number (105.404 mg/g) since it is highly unlikely that the value is precise to three decimal digits. If possible, include the uncertainty of the value.

Response 2: Thanks for your comments. This sentence has been changed in the revised manuscript.

Point 3: Lines 19-21: you present here contradictory information, since you refer as optimal dose of peat both 3200 mg/L and 3.6 g/L. I understand that the latter is provided by the orthogonal design, but you cannot present both results as the best dosage since it is inaccurate.

Response 3: Thanks for your comments. This contradictory information has been deleted in the revised manuscript avoid misunderstanding.

Introduction

Point 4: Line 27: instead of "the least neglected" write "one of the most important"

Response 4: Thanks for your comments. "the least neglected" has been changed to "one of the most important" in the revised manuscript.

Point 5: Line 28: instead of "industrial activities such as electroplating" write "industrial activities, from electroplating"

Response 5: Thanks for your comments. "industrial activities such as electroplating" has been changed to "industrial activities, from electroplating" in the revised manuscript.

Point 6: Line 31: instead of "with the help of rainwater of which Cr(III) and Cr(VI)" write "with the help of rain water. Cr(III) and Cr(VI)"

Response 6: Thanks for your comments. "with the help of rainwater of which Cr(III) and Cr(VI)" has been changed to "with the help of rain water. Cr(III) and Cr(VI)" in the revised manuscript.

Point 7: Line 36: instead of "increased attentions have been put on its damages" write "increased attention has been drawn to its damages"

Response 7: Thanks for your comments. "increased attentions have been put on its damages" has been changed to "increased attention has been drawn to its damages" in the revised manuscript.

Point 8: Line 38: instead of "reported the capability" write "reported on the capability"

Response 8: Thanks for your comments. "reported the capability" has been changed to "reported on the capability" in the revised manuscript.

Point 9: Line 41: instead of "many researches" write "many studies"

Response 9: Thanks for your comments. "many researches" has been changed to "many studies" in the revised manuscript.

Point 10: Line 43: instead of "due to high cost" write "due to their high cost"

Response 10: Thanks for your comments. "due to high cost" has been changed to "due to their high cost" in the revised manuscript.

Point 11: Line 43: instead of "recent researches" write "recent studies"

Response 11: Thanks for your comments. "recent researches" has been changed to "recent studies" in the revised manuscript.

Point 12: Line 48: instead of "contains large quantity" write "contains a large quantity"

Response 12: Thanks for your comments. "contains large quantity" has been changed to "contains a large quantity" in the revised manuscript.

Point 13: Lines 48-49: instead of "humic substances, and has a large" write "humic substances, has a large"

Response 13: Thanks for your comments. "humic substances, and has a large" has been changed to "humic substances, has a large" in the revised manuscript.

Point 14: Line 53: instead of "are traced back" write "can be traced back"

Response 14: Thanks for your comments. "are traced back" has been changed to "can be traced back" in the revised manuscript.

Point 15: Line 54: instead of "researches" write "studies"

Response 15: Thanks for your comments. "researches" has been changed to "studies" in the revised manuscript.

Point 16: Line 57: instead of "have been examined" write "have been examining"

Response 16: Thanks for your comments. "have been examined" has been changed to "have been examining" in the revised manuscript.

Point 17: Line 60: instead of "as the adsorbent" write "as adsorbent"

Response 17: Thanks for your comments. "as the adsorbent" has been changed to "as adsorbent" in the revised manuscript.

Point 18: Line 64: instead of "when used" write "when it is used"

Response 18: Thanks for your comments. "when used" has been changed to "when it is used" in the revised manuscript.

Point 19: Line 68: instead of "following 4.9 ± 0.01 mg/g of peat" write "followed by 4.90 ± 0.01 mg/g in peat"

Response 19: Thanks for your comments. "following 4.9 ± 0.01 mg/g of peat" has been changed to "followed by 4.90 ± 0.01 mg/g in peat" in the revised manuscript.

Point 20: Line 69: instead of "18.6 mg/g of peat" write "18.6 mg/g in peat"

Response 20: Thanks for your comments. "18.6 mg/g of peat" has been changed to "18.6 mg/g in peat" in the revised manuscript.

Point 21: Line 73: instead of "increased attentions are focused" write "further research is being focused"

Response 21: Thanks for your comments. "increased attentions are focused" has been changed to "further research is being focused" in the revised manuscript.

Point 22: Lines 74-75: This has been mentioned in the previous review, this sentence is contradictory with what was presented earlier in the paragraph. I would review or remove it.

Response 22: Thanks for your comments. This sentence has been deleted in the revised manuscript.

Point 23: Line 80: instead of "kinetics characteristics were evaluated. The interaction mechanism of Cr(VI) to" write "kinetics and the interaction mechanism of Cr(VI) with"

Response 23: Thanks for your comments. "kinetics characteristics were evaluated. The interaction mechanism of Cr(VI) to" has been changed to "kinetics and the interaction mechanism of Cr(VI) with" in the revised manuscript.

Point 24: Line 81: instead of "was also examined" write "were also examined"

Response 24: Thanks for your comments. "was also examined" has been changed to "were also examined" in the revised manuscript.

Materials and Methods

Point 25: Line 94: instead of "were then tested, whereupon the best is identified", write "was then tested, in order to identify the best performing one".

Response 25: Thanks for your comments. "were then tested, whereupon the best is identified" has been changed to "was then tested, in order to identify the best performing one" in the revised manuscript.

Point 26: Line 111: instead of "using FT-IR and by analyzing changes" write "using FT-IR by analyzing changes"

Response 26: Thanks for your comments. "using FT-IR and by analyzing changes" has been changed to "using FT-IR by analyzing changes" in the revised manuscript.

Point 27: Line 114: please review the significant digits according to the uncertainty. If the uncertainty has three decimal digits then the value should have as well, e.g. 0.030 ± 0.002 g

Response 27: Thanks for your comments. This error, as well as similar errors in the manuscript, has been changed in the revised manuscript.

Point 28: Lines 114-117: refer the pH at which these experiments were carried out

Response 28: Thanks for your comments. The experiment condition of pH has been added in the revised manuscript.

Point 29: Line 118: instead of "adsorbing capacity", write "adsorbed amount". Also, the terminology for adsorbed amount is usually "q" and not "Y".

Response 29: Thanks for your comments. "adsorbing capacity" has been changed to "adsorbed amount" in the revised manuscript. "Y" has been changed to "q" in the revised manuscript.

Point 30: Line 118-119: eliminate "of which the experimental data was fitted using the Langmuir and Freundlich isotherm models", this is redundant, since it will be explained later in the text.

Response 30: Thanks for your comments. This sentence has been eliminated in the revised manuscript.

Point 31: Line 121: instead of "after equilibrium" write "at equilibrium"

Response 31: Thanks for your comments. "after equilibrium" has been changed to "at equilibrium" in the revised manuscript.

Point 32: Line 122: instead of "chromium" write "solution"

Response 32: Thanks for your comments. "chromium" has been changed to "solution" in the revised manuscript.

Point 33: Line 122: instead of "mL" write "L"

Response 33: Thanks for your comments. "mL" has been changed to "L" in the revised manuscript.

Point 34: Line 127: same with significant digits as in line 114. Also, explain why a different dosage was used for isotherms and for kinetics?

Response 34: Thanks for your comments. The experiment condition of pH has been added in the revised manuscript. Peat dose has no effect on the fitting results, but in order to avoid misunderstanding, delete this experiment condition here.

Point 35: Lines 127-131: refer the pH at which these experiments were carried out

Response 35: Thanks for your comments. The experiment condition of pH has been added in the revised manuscript.

Point 36: Line 132: instead of "the adsorbances of" write "the adsorbed amounts for"

Response 36: Thanks for your comments. "the adsorbances of" has been changed to "the adsorbed amounts for" in the revised manuscript.

Point 37: Lines 133-134: instead of "a pseudo-second-order kinetic model" write "pseudo-second-order kinetic model"

Response 37: Thanks for your comments. "a pseudo-second-order kinetic model" has been changed to "pseudo-second-order kinetic model" in the revised manuscript.

Point 38: Line 140: instead of "amounts of 0.5 mol/L HNO3 and 0.5 mol/L NaOH" write "amount of 0.5 mol/L HNO3 or 0.5 mol/L NaOH"

Response 38: Thanks for your comments. "amounts of 0.5 mol/L HNO3 and 0.5 mol/L NaOH" has been changed to "amount of 0.5 mol/L HNO3 or 0.5 mol/L NaOH" in the revised manuscript.

Point 39: Line 143: be more precise with the parameters, "about 3 and 4" is very vague.

Response 39: Thanks for your comments. This parameter has been changed in the revised manuscript.

Point 40: Lines 140-146: explain more precisely the single factor experiments step by step for each factor, including the fixed parameters when one of the factors was varied.

Response 40: Thanks for your comments. The experimental conditions step by step in the single factor experiments have been added in the revised manuscript.

Point 41: Line 148: instead of "with 4-factor, 4-level" write "with 4 factors and 4 levels"

Response 41: Thanks for your comments. "with 4-factor, 4-level" has been changed to "with 4 factors and 4 levels" in the revised manuscript.

Point 42: Line 148: I cannot find reference [39], and it does not support the theory of the experimental design. Please review

Response 42: Thanks for your comments. The previous reference cited was a Chinese journal, now we have re-quoted a new reference in the revised manuscript.

Point 43: Line 148: instead of "The 16 above samples", write "The 16 samples in the conditions of the experimental design"

Response 43: Thanks for your comments. "The 16 above samples" has been changed to "The 16 samples in the conditions of the experimental design" in the revised manuscript.

Results and Discussion

Point 44: Line 158: instead of "stock solutions thereby generationg oxygen" write "stock solutions. This means that oxygen"

Response 44: Thanks for your comments. "stock solutions thereby generationg oxygen" has been changed to "stock solutions. This means that oxygen" in the revised manuscript.

Point 45: Line 159: instead of "and creating active sites" write "were generated creating active sites"

Response 45: Thanks for your comments. "and creating active sites" has been changed to "were generated creating active sites" in the revised manuscript.

Point 46: Line 161: instead of "sulfonate" write "sulfonation"

Response 46: Thanks for your comments. "sulfonate" has been changed to "sulfonation" in the revised manuscript.

Point 47: Line 161: instead of "in a low" write "in lower"

Response 47: Thanks for your comments. "in a low" has been changed to "in lower" in the revised manuscript.

Point 48: Line 173: instead of "and a peat dose" write "peat dose"

Response 48: Thanks for your comments. "and a peat dose" has been changed to "peat dose" in the revised manuscript.

Point 49: Line 178: instead of "1500 times magnification" write "1500 times"

Response 49: Thanks for your comments. "1500 times magnification" has been changed to "1500 times" in the revised manuscript.

Point 50: Line 181: instead of "between its aggregate" write "between its aggregates"

Response 50: Thanks for your comments. "between its aggregate" has been changed to "between its aggregates" in the revised manuscript.

Point 51: Lines 183-187: I cannot agree with the observations. The imagines do not show the fine porosity (it is nm-sized), only macropores (μm-sized), and the natural peat actually seems to show very little porosity. Similarly, the images do not have enough resolution for conclusions to be drawn about the surface smoothness. This explanation needs to be reviewed.

Response 51: Thanks for your comments. This explanation has been changed in the revised manuscript.

Point 52: Line 191: instead of "are demonstrated", write "are shown"

Response 52: Thanks for your comments. "are demonstrated" has been changed to "are shown" in the revised manuscript.

Point 53: Line 191: eliminate "The result is beyond all expectations", this is speculative

Response 53: Thanks for your comments. "The result is beyond all expectations" has been deleted in the revised manuscript.

Point 54: Line 193: instead of "have slackened" write "have decreased"

Response 54: Thanks for your comments. "have slackened" has been changed to "have decreased" in the revised manuscript.

Point 55: Line 204: instead of "adsorbance peat" write "Cr-loaded peat"

Response 55: Thanks for your comments. "adsorbance peat" has been changed to "Cr-loaded peat" in the revised manuscript.

Point 56: Table 2: review the significant numbers, I doubt that the numbers are significant to the fourth decimal digit or the seventh decimal digit in the case of pore volume. Also, if possible, include the uncertainties of the determinations.

Response 56: Thanks for your comments. This problem has been changed in the revised manuscript.

Point 57: Line 207: instead of "was basically stronger" write "was stronger"

Response 57: Thanks for your comments. "was basically stronger" has been changed to "was stronger" in the revised manuscript.

Point 58: Lines 210-213: In my opinion, peaks at 1699 cm-1 and 1503 cm-1 were misidentified, since they are not important features in the spectrum through the observation in the graph of Fig. 3. I’m not sure if it’s relevant for them to be mentioned in the text.

Response 58: Thanks for your comments. After reanalysis, it is true that the peaks of the two places are not obvious, so we decided to delete the relevant description to avoid misunderstanding.

Point 59: Line 217: On the other hand, the peak at 1035 cm-1 is not referred and is quite significant for all the acquired spectra.

Response 59: Thanks for your comments. The peak at 1035 cm-1 is very important and referred in the revised manuscript.

Point 60: Lines 220-221: The statement that 1035 cm-1 is a new vibration peak is inaccurate, since in Fig. 3 a peak can be seen in all spectra at this wave number, even though the shape is different. On the other hand, the peak at 1169 cm-1 is only present in the modified peat samples, and this is not referred in the text.

Response 60: Thanks for your comments. This analysis has been changed in the revised manuscript.

Point 61: Lines 224-225: I’m not sure I agree that peaks at 1035 cm-1, 795 cm-1, and 630 cm-1 are weaker in Cr-modified peat; the only relevant difference is seen for the 1169 cm-1 peak, in my opinion.

Response 61: Thanks for your comments. This analysis has been changed in the revised manuscript.

Point 62: Lines 230-232: How are these conclusions drawn? FTIR is not quantitative, and provides no information about the surface acidity, hydrophilicity, chelation and ion exchange properties. The only inference that could possibly be said is that due to the magnitude of the peaks, it is likely that modified peat has more functional groups available for adsorption than natural peat. Other statements are speculative.

Response 62: Thanks for your comments. This analysis has been changed in the revised manuscript.

Point 63: Figure 3: in the legend, write "natural peat" instead of "nature peat"

Response 63: Thanks for your comments. "nature peat" has been changed to "natural peat" in the revised manuscript.

Point 64: Line 237: instead of "the interaction" write "the equilibrium"

Response 64: Thanks for your comments. "the interaction" has been changed to "the equilibrium" in the revised manuscript.

Point 65: Figure 4: y-axis legend should be "adsorbed amount" instead of "adsorbance", and mg/g seems to be the more appropriate unit, given the order of magnitude of the values.

Response 65: Thanks for your comments. "adsorbance" has been changed to "adsorbed amount" in the revised manuscript. The unit has been changed in the revised manuscript.

Point 66: Line 241: instead of "in the Langmuir" write "with the Langmuir"

Response 66: Thanks for your comments. "in the Langmuir" has been changed to "with the Langmuir" in the revised manuscript.

Point 67: Lines 243-244: This sentence describes a normal behavior, in my opinion it is redundant to refer to this in the text

Response 67: Thanks for your comments. This sentence has been deleted in the revised manuscript.

Point 68: Lines 244-248: Very confusing statements, need thorough revision or removal from the text

Response 68: Thanks for your comments. This statement has been deleted in the revised manuscript.

Point 69: Line 248: instead of "As a result, many adsorption models" write "Many adsorption models"

Response 69: Thanks for your comments. "As a result, many adsorption models" has been changed to "Many adsorption models" in the revised manuscript.

Point 70: Line 258: instead of "adsorption thermodynamic processes of" write "adsorption processes in"

Response 70: Thanks for your comments. "adsorption thermodynamic processes of" has been changed to "adsorption processes in" in the revised manuscript.

Point 71: Lines 258-261: again very confusing statements that need thorough revision or removal from the text

Response 71: Thanks for your comments. This sentence has been changed in the revised manuscript.

Point 72: Line 263: units of KF should be (mg g-1) (L mg-1)1/n

Response 72: Thanks for your comments. This unit has been changed in the revised manuscript.

Point 73: Table 3: review the significant numbers of the presented values, highly unlikely that they are relevant to the fifth decimal digit. If possible, provide uncertainty of the values of the parameters. Refer to the units of KF

Response 73: Thanks for your comments. We are going to leave the significant numbers a little bit smaller to increase the uncertainty of the data. But there is no concrete uncertainty in the computer model fitting.

Point 74: Lines 268-269: instead of "The adsorption isotherm of the modified peat was fitted to the Langmuir and Freundlich adsorption isothermal models, of which the fitting results are presented" write "The Langmuir and Freundlich adsorption isotherm models were fitted to the experimental data, and the fitting results are presented"

Response 74: Thanks for your comments. "The adsorption isotherm of the modified peat was fitted to the Langmuir and Freundlich adsorption isothermal models, of which the fitting results are presented" has been changed to "The Langmuir and Freundlich adsorption isotherm models were fitted to the experimental data, and the fitting results are presented" in the revised manuscript.

Point 75: Lines 269-270: instead of "The adsorption of Cr(VI) by the modified peat can be fitted" write "Good fitting can be achieved"

Response 75: Thanks for your comments. "The adsorption of Cr(VI) by the modified peat can be fitted" has been changed to "Good fitting can be achieved" in the revised manuscript.

Point 76: Line 272: instead of "the active points on the peat surface decrease" write "the decrease in available adsorption sites at the surface"

Response 76: Thanks for your comments. "the active points on the peat surface decrease" has been changed to "the decrease in available adsorption sites at the surface" in the revised manuscript.

Point 77: Lines 272-273: clarify the meaning of "adsorbate enter the granular layers" (confusing statement)

Response 77: Thanks for your comments. We changed the confusing words in the revised manuscript.

Point 78: Line 274: instead of "to reach as high as" write "to reach"

Response 78: Thanks for your comments. "to reach as high as" has been changed to "to reach" in the revised manuscript.

Point 79: Lines 274-277: the information presented for comparison is very incomplete. Which material and conditions were used in these studies?

Response 79: Thanks for your comments. The information presented for comparison has been changed in the revised manuscript.

Point 80: Line 281: "a progressively expedited pace" – this is incorrect, actually the rate of adsorption is fast at the beginning and then slows down, please review the statement

Response 80: Thanks for your comments. The expression has been changed in the revised manuscript.

Point 81: Line 284: instead of "is presented" write "are presented"

Response 81: Thanks for your comments. "is presented" has been changed to "are presented" in the revised manuscript.

Point 82: Lines 284-285: instead of "the experimental results fit the pseudo-second-order kinetic model" write "the pseudo-second-order model fits well the experimental results"

Response 82: Thanks for your comments. "the experimental results fit the pseudo-second-order kinetic model" has been changed to "the pseudo-second-order model fits well the experimental results" in the revised manuscript.

Point 83: Line 287: instead of "kinetic plots follows the Equation 4" write "kinetic plot follows Equation 4"

Response 83: Thanks for your comments. "kinetic plots follows the Equation 4" has been changed to "kinetic plot follows Equation 4" in the revised manuscript.

Point 84: Line 288: instead of "deduced to" write "transformed into"

Response 84: Thanks for your comments. "deduced to" has been changed to "transformed into" in the revised manuscript.

Point 85: Line 289: instead of "kinetic plots follows the Equation 6" write "kinetic plot follows Equation 6"

Response 85: Thanks for your comments. "kinetic plots follows the Equation 6" has been changed to "kinetic plot follows Equation 6" in the revised manuscript.

Point 86: Line 290: instead of "deduced to" write "transformed into"

Response 86: Thanks for your comments. "deduced to" has been changed to "transformed into" in the revised manuscript.

Point 87: Line 291: instead of "In these Equation, Qe is the equilibrium maximum adsorbance (mg/kg)" write "In these equations, Qe is the adsorbed amount at equilibrium (mg/g)"

Response 87: Thanks for your comments. "In these Equation, Qe is the equilibrium maximum adsorbance (mg/kg)" has been changed to "In these equations, Qe is the adsorbed amount at equilibrium (mg/g)" in the revised manuscript.

Point 88: Lines 291-292: refer to the difference between k1 (pseudo-first-order rate) and k2 (pseudo-second-order rate)

Response 88: Thanks for your comments. The difference between k1 and k2 is added in the revised manuscript.

Point 89: Lines 292-293: instead of "the adsorbance in a specific period (mg/kg)" write "the adsorbed amount at a specific time t (mg/g)"

Response 89: Thanks for your comments. "the adsorbance in a specific period (mg/kg)" has been changed to "the adsorbed amount at a specific time t (mg/g)" in the revised manuscript.

Point 90: Line 294: instead of "follows the Equation 8" write "follows Equation 8"

Response 90: Thanks for your comments. "follows the Equation 8" has been changed to "follows Equation 8" in the revised manuscript.

Point 91: Line 295: describe better the meaning of α and β coefficients and refer to their units

Response 91: Thanks for your comments. Here α and β are merely coefficients, with no specific meaning or units.

Point 92: Lines 295-296: instead of "the adsorbance in a specific period (mg/kg)" write "the adsorbed amount at a specific time t (mg/g)"

Response 92: Thanks for your comments. "the adsorbance in a specific period (mg/kg)" has been changed to "the adsorbed amount at a specific time t (mg/g)" in the revised manuscript.

Point 93: Line 296: instead of "the equilibrium adsorbance (mg/kg)" write "the adsorbed amount at equilibrium (mg/g)"

Response 93: Thanks for your comments. "the equilibrium adsorbance (mg/kg)" has been changed to "the adsorbed amount at equilibrium (mg/g)" in the revised manuscript.

Point 94: Line 297: eliminate "is presented"

Response 94: Thanks for your comments. "is presented" has been deleted in the revised manuscript.

Point 95: Line 298: instead of "are demonstrated" write "are presented"

Response 95: Thanks for your comments. "are demonstrated" has been changed to "are presented" in the revised manuscript.

Point 96: Figure 5: y-axis of graph A : instead of "adsorbance (mg/kg)" it should be "adsorbed amount (mg/g)"

Response 96: Thanks for your comments. "adsorbance (mg/kg)" has been changed to "adsorbed amount (mg/g)" in the revised manuscript.

Point 97: Line 300: instead of "Adsorption kinetics (A)" write "Adsorption kinetics with (A)"

Response 97: Thanks for your comments. "Adsorption kinetics (A)" has been changed to "Adsorption kinetics with (A)" in the revised manuscript.

Point 98: Table 4: there are parameters missing in this Table, it is incomplete

Response 98: Thanks for your comments. The parameters obtained after the transformation into linear fitting are only slope and fitting degree. The value of R2 of the fitting degree is used to determine which model has a better fitting degree.

Point 99: Line 306-307: "thereby revealing the adsorption mechanism" – a speculative statement, I suggest eliminating

Response 99: Thanks for your comments. This sentence has been deleted in the revised manuscript.

Point 100: Lines 308-309: instead of "the chemisorption of the valence forces or covalent forces between the adsorbent and the adsorbate" write "chemisorption"

Response 100: Thanks for your comments. "the chemisorption of the valence forces or covalent forces between the adsorbent and the adsorbate" has been changed to "chemisorption" in the revised manuscript.

Point 101: Lines 309-310: instead of "the electronic sharing" write "electron sharing"

Response 101: Thanks for your comments. "the electronic sharing" has been changed to "electron sharing" in the revised manuscript.

Point 102: Line 313: instead of "adsorption process" write "adsorption kinetics"

Response 102: Thanks for your comments. "adsorption process" has been changed to "adsorption kinetics" in the revised manuscript.

Point 103: Lines 312-315: very confusing statements, need revision or removal

Response 103: Thanks for your comments.This sentence has been changed in the revised manuscript.

Point 104: Line 323: instead of "was characterized, of which the results" write "was evaluated and the results"

Response 104: Thanks for your comments. "was characterized, of which the results" has been changed to "was evaluated and the results" in the revised manuscript.

Point 105: Line 324: instead of "kept accelerating" write "kept increasing"

Response 105: Thanks for your comments. "kept accelerating" has been changed to "kept increasing" in the revised manuscript.

Point 106: Line 324: "0.08 g" – please present the dosage as concentrations instead of absolute values, for better comparison with other studies and better understanding. The same in the axis of Fig. 6.

Response 106: Thanks for your comments. The unit of peat dose has been changed in the revised manuscript.

Point 107: Line 331: instead of "sped up in tandem with the rise" write "increased slightly with the rise"

Response 107: Thanks for your comments. "sped up in tandem with the rise" has been changed to "increased slightly with the rise" in the revised manuscript.

Point 108: Line 332: instead of "if conditions permit" write "if conditions allow"

Response 108: Thanks for your comments. "if conditions permit" has been changed to "if conditions allow" in the revised manuscript.

Point 109: Line 333: instead of "faster adsorption" write "enhanced adsorption"

Response 109: Thanks for your comments. "faster adsorption" has been changed to "enhanced adsorption" in the revised manuscript.

Point 110: Line 338: instead of "decelerates" write "decreases"

Response 110: Thanks for your comments. "decelerates" has been changed to "decreases" in the revised manuscript.

Point 111: Line 339: instead of "the deceleration slows down" write "the decrease stabilizes"

Response 111: Thanks for your comments. "the deceleration slows down" has been changed to "the decrease stabilizes" in the revised manuscript.

Point 112: Line 339-340: not sure what you mean by this sentence. I suggest it is removed.

Response 112: Thanks for your comments.This sentence has been removed in the revised manuscript.

Point 113: Line 341: instead of "wherein" write "and they show that"

Response 113: Thanks for your comments. "wherein" has been changed to "and they show that" in the revised manuscript.

Point 114: Lines 342-344: how is this conclusion drawn from the data? Also, I saw nothing in ref. 5 to support this statement. Please revise.

Response 114: Thanks for your comments. This sentence has been changed in the revised manuscript.

Point 115: Lines 344-345: please revise this statement since surface neutralization cannot be occurring, because Cr(VI) is anionic.

Response 115: Thanks for your comments. This sentence has been changed in the revised manuscript.

Point 116: Lines 346-348: This statement also needs revision since Cr6+ has reductive and not oxidative potential.

Response 116: Thanks for your comments. This sentence has been changed in the revised manuscript.

Point 117: Lines 350-351: instead of "diffused to their reaction positions" write "diffused"

Response 117: Thanks for your comments. "diffused to their reaction positions" has been changed to "diffused" in the revised manuscript.

Point 118: Line 353: "the rise in pH was promoted by ion exchange" – nothing in the results suggest this, please explain or revise the statement

Response 118: Thanks for your comments. This sentence has been changed in the revised manuscript.

Point 119: Line 354: instead of "Cr6+" it should be "Cr3+"

Response 119: Thanks for your comments. "Cr6+" has been changed to "Cr3+" in the revised manuscript.

Point 120: Lines 356-359: this statement also needs revision since it indicates that Cr removal is favoured at higher pH conditions and this is contradictory to the experimental results.

Response 120: Thanks for your comments. In fact, what is explained here is the possible reaction of the pH to the removal of trivalent chromium. Explain the subsequent reaction that may occur when hexavalent chromium is converted to trivalent chromium, which is the formation of precipitation.

Point 121: Table 5: the meaning of the K1, K2, K3, K4 and R parameters was not clarified.

Response 121: Thanks for your comments. K represents the mean value of a certain level of a factor, and R represents the poor value of K. K1 represents the average removal efficiency at first level of each influencing factor. K2 represents the average removal efficiency at second level of each influencing factor. K3 represents the average removal efficiency at third level of each influencing factor. K4 represents the average removal efficiency at forth level of each influencing factor.

Point 122: Table 6: Sum of squares values are very high, please revise your results, also instead of "degree of freedom" write "degrees of freedom"

Response 122: Thanks for your comments. This result is automatically calculated by SPSS, and we have verified it. But for the sake of aesthetics, we choose to keep two significant digits.

Point 123: Line 369: instead of "the significant level" write "with a significance level"

Response 123: Thanks for your comments. "the significant level" has been changed to "with a significance level" in the revised manuscript.

Point 124: Line 369: instead of "9.28, F" write "9.28. This means that since F"

Response 124: Thanks for your comments. "9.28, F" has been changed to "9.28. This means that since F" in the revised manuscript.

Point 125: Line 370: instead of "indicating" write "this indicates"

Response 125: Thanks for your comments. "indicating" has been changed to "this indicates" in the revised manuscript.

Point 126: Line 371: instead of "F(temperature) < 9.28, indicating that temperature" write "while since F(temperature) < 9.28, temperature"

Response 126: Thanks for your comments. "F(temperature) < 9.28, indicating that temperature" has been changed to "while since F(temperature) < 9.28, temperature" in the revised manuscript.

Point 127: Line 372: instead of "sulfonated peat, indicating that pH" write "sulfonated peat. Moreover, F(pH) > 9.28, indicating that pH"

Response 127: Thanks for your comments. "sulfonated peat, indicating that pH" has been changed to "sulfonated peat. Moreover, F(pH) > 9.28, indicating that pH" in the revised manuscript.

Point 128: Line 373: instead of "sulfonated peat, F(addition)> 9.28" write "sulfonated peat, and F(peat dose) > 9.28"

Response 128: Thanks for your comments. "sulfonated peat, F(addition)> 9.28" has been changed to "sulfonated peat, and F(peat dose) > 9.28" in the revised manuscript.

Point 129: Line 374: instead of "the addition of peat" write "the peat dose also"

Response 129: Thanks for your comments. "the addition of peat" has been changed to "the peat dose also" in the revised manuscript.

Point 130: Lines 375-376: the optimal conditions here presented are not consistent with the single factor experiments. You refer this for temperature on lines 376-378, but this is also valid for the other factors, and it is not explained well.

Response 130: Thanks for your comments. We have deleted vague statements to avoid confusion. Orthogonal experiment is mainly based on single factor experiment in order to judge the influence degree of influencing factors and optimize conditions

Point 131: Line 380: instead of "their strengths", write "their influence"

Response 131: Thanks for your comments. "their strengths" has been changed to "their influence" in the revised manuscript.

Point 132: Lines 380-381: please explain the sorting of the influencing factors, since it does not seem to be consistent with F or significance values.

Response 132: Thanks for your comments. The ranking of influence degree of influencing factors is mainly conducted according to R value. In anova, the larger the F statistic is, the more it indicates that inter-group variance is the main source of variance.

Point 133: Lines 381-384: these statements are very confusing and need to be revised.

Response 133: Thanks for your comments. This sentence has been changed in the revised manuscript.  

 Conclusions

Point 134: Line 390: "under the presented conditions" – clarify and describe these conditions

Response 134: Thanks for your comments. The "presented conditions " here refers to all the conditions for thermodynamic experiments.

Point 135: Lines 390-391: instead of "The kinetic equation of the adsorption process fit the pseudo-second-order kinetic model" write "The pseudo-second-order model fit the adsorption kinetics"

Response 135: Thanks for your comments. "The kinetic equation of the adsorption process fit the pseudo-second-order kinetic model" has been changed to "The pseudo-second-order model fit the adsorption kinetics" in the revised manuscript.

Point 136: Lines 392-393: instead of "conducted isothermal adsorption experiment, the adsorption thermodynamic process" write "conducted adsorption experiments, the adsorption process"

Response 136: Thanks for your comments. "conducted isothermal adsorption experiment, the adsorption thermodynamic process" has been changed to "conducted adsorption experiments, the adsorption process" in the revised manuscript.

Point 137: Line 393: instead of "Freundlich isothermal" write "Freundlich isotherm"

Response 137: Thanks for your comments. "Freundlich isothermal" has been changed to "Freundlich isotherm" in the revised manuscript.

Point 138: Line 396: instead of "single factor experiment and orthogonal experiment" write "single factor experiments and orthogonal experiments"

Response 138: Thanks for your comments. "single factor experiment and orthogonal experiment" has been changed to "single factor experiments and orthogonal experiments" in the revised manuscript.

Point 139: Lines 396-398: these conclusions need to be revised according to the revision required in the results and discussion section.

Response 139: Thanks for your comments. This conclusion has been changed in the revised manuscript.

Reviewer 2 Report

*Major Points*

*1.**Line 86 to 89: “Three different treatments were applied: Firstly,
the peat was steeped in H_2 SO_4 for 24 h at room temperature; secondly,
the peat was steeped in H_2 SO_4 and heated for 2 h in a water bath at
90 ºC then steeped 24 h at room temperature; and lastly, the peat was
steeped in H_2 SO_4 and heated for 2 h in an electric cooker at 600 ºC
then steeped 24 h at room temperature.” ***

There are three different experiments with three different experimental
conditions but words such as “firstly”, “secondly” and“lastly” are
confusing andgive the appearance ofconsecutive steps in only one
experiment. Each experiment should be mentioned separately in its own
sentence.

*2.**Line 84 to 92: *The approximate amount of peat and sulfuric acid
used in each treatment should be mentioned. The “solid-to-liquid ratio”
does not give enough information to repeat the experiment. **

**

*3.**Line 98: “and then oscillated for 4 h at room temperature at a
speed of 300 rpm.” *

What equipment was used for this?

*4.**Lines 123 to 124: “The amount of metal adsorbed on peat served as a
metric for the concentration change between the initial solution and the
measured filtrate” *

This sentence does not make sense. The order in the sentence should be
changed.

*5.**Lines137 to 140: “The 50 mg/L Cr(VI) solutions were partitioned
into 25 mL portions with different pHs and were transferred separately
into centrifuge tubes with 0.08 g of modified peat in each tube. The pH
of the solutions were adjusted to the following values using an
appropriate amounts of 0.5 mol/L HNO3 and 0.5 mol/L NaOH: 1.5, 2, 3, 4,
5, 6, 7, 8, 9, 10, and 11.”*

Was the pH adjusted after of the partition? This could change the volume
and therefore the

concentration of metal.

**

*6.**Line 147 to 151:* *“To assess the influence of the factors on the
adsorption of the samples, an L16(44) orthogonal experiment with
4-factor, 4-level was designed [39], which is shown in Table 1. The 16
above samples were oscillated for 4 h at a rate of 300 rpm before they
were centrifuged. Their chromium ion concentrations were then measured
at equilibrium. The data were analyzed with SPSS statistics (IBM 150 21,
USA).”*

The manuscript should explain in simple words the meaning and usefulness
of this statistic analysis.

*7.**Line 241 and 242: “Figure 4. Adsorption isotherms of the modified
peat in the Langmuir (red) and Freundlich (blue) models.”*

The figure 4 should be placed after of the explanation of the Langmuir
and Freundlich model.

*8.**Line 267:* *Table 3. Isotherm parameters*

*Two points:*

1)The used units for the Langmuir model in the table for the maximum
amount adsorbed predicted by the model(q_max ) are mg/g . The units used
in the explanation of the Langmuir model in the manuscript (Line 264)
and the Figure 4 are mg/kg. What was the reason of the inconsistency in
the units? What unit was used for the calculation of q_max ?

2)How the value of q_max of Langmuir was calculated? Was a linear
equation form of Langmuir used? If the answer is yes, the linear
equation should be included in the explanation of the model.

*9.**Lines 274 to 277: “According to Kołoczek et al. [32], theadsorption
capacity of Cr(VI) (adsorption capacity of 18.75 mg/g) was significantly
inferior to thevalues presented in this study. A research through column
experiment concluded that the best Cr(VI) adsorption capacity can reach
65.87 mg/g, which currently cannot be achieved with natural peat [33].” *

These sentences should be rewritten because they are confusing,
especially the last one.

*10.**Line 280: “The adsorption properties were studied in a dynamic
regime.” *

What is the meaning of this sentence?

*11.**Lines 291 and 304:“In these Equation, Qe is the equilibrium
maximum adsorbance (mg/kg)”and “According to Fig. 5A, the adsorption
capacity is approximately 22mg/g”*

Two points:

1)Line 291, What is the usefulness of using mg/kg as units instead of
mg/g? The last one is the usual unit seen in the literature and three
zeros could be avoided .

2)Line 304, the authors are not consistent in the units.

*12.**Lines 302 and 303: Table 4. Parameters of the linear forms of the
pseudo-first adsorption kinetic, pseudo-secondadsorption kinetic, and
Elovich models.*

It is not possible for k to be negative, the authors are wrong in the
interpretation of the pseudo-first order adsorption kinetic equation and
how to obtain the value of the rate constant.If a reaction is of first
order, a linear plot has a slope of -k.

*13.**Lines 328 to 333: “Figure 6. Effects of the adsorbent dose (A),
temperature (B), and pH (C) on the removal of Cr(VI) by the modified peat.”*

The author concludes that the adsorption is an endothermic process based
on Figure 6 B but the difference in percent of removal of metal due to
the temperature between 10℃and 50℃is not significant. Likewise there are
not correlation between the increase in temperature and the increase in
percent of removal of metal at some points. The percent of removal of
metal falls at the temperatures of 20℃and 40℃. How these results support
the idea of an endothermic process?

*14.**Lines 346 to 349:* *“3C+ 2Cr_2 O_7 ^2- + 16H^+ →4Cr^3+ + 3CO_2 +
8H_2 O”*

What is the meaning of 3C? The phrase “C represents the organic matter
in this experiment” is a too broad definition and it is not enough to
understand how the authors obtained the coefficients of the
equation.Even 3C could be seen as atomic Carbon.

*15.**Line 356 to 361*

Two points:

1)This paragraph is confusing. What are the authors trying to say in the
paragraph?

2)The authors recognize the possibility of precipitation at higher
levels of pH but they do not mention the consequences of this
precipitation in the results of the current experiment.

*Minor points:*

*1.Line 27 : “chrome residues are the least neglected”*

What is the meaning of the phrase?

**

*2.Line 36 and 73: “increased attentions”*

The plural form is not correct.

**

*3.Line 74: “some modification method do not present”*

The word should be “methods”.

**

*4.Line 85: *The source of concentrated sulfuric acid.**

*5.**Line 94: “whereupon the best is identified.”*

The word should be “was”.

*6.Line 95: *The source of potassium dichromate (K_2 Cr_2 O_7 ).

Author Response

Response to Reviewer 2 Comments

Major Points

Point 1: Line 86 to 89: "Three different treatments were applied: Firstly, the peat was steeped in H2SO4 for 24 h at room temperature; secondly, the peat was steeped in H2SO4and heated for 2 h in a water  bath at 90 ºC then steeped 24 h at room temperature; and lastly, the peat was steeped in H2SO4and  heated for 2 h in an electric cooker at 600 ºC then steeped 24 h at room temperature."  

There are three different experiments with three different experimental conditions but words such as "firstly", "secondly" and  "lastly" are confusing and  give the appearance of  consecutive steps in only one experiment. Each experiment should be mentioned separately in its own sentence.

Response 1: Thanks for your comments. This sentence has been changed in the revised manuscript.

Point 2: Line 84 to 92: The approximate amount of peat and sulfuric acid used in each treatment should be mentioned. The "solid-to-liquid ratio" does not give enough information to repeat the experiment.

Response 2: Thanks for your comments. This information has been added in the revised manuscript. 

Point 3: Line 98: "and then oscillated for 4 h at room temperature at a speed of 300 rpm."

What equipment was used for this?

Response 3: Thanks for your comments. This information has been added in the revised manuscript.

Point 4: Lines 123 to 124: "The amount of metal adsorbed on peat served as a metric for the concentration change between the initial solution and the measured filtrate"

This sentence does not make sense. The order in the sentence should be changed.  

Response 4: Thanks for your comments. This sentence has been moved to the front in the revised manuscript.

Point 5: Lines 137 to 140: "The 50 mg/L Cr(VI) solutions were partitioned into 25 mL portions with different pHs and were transferred separately into centrifuge tubes with 0.08 g of modified peat in each tube. The pH of the solutions were adjusted to the following values using an appropriate amounts of 0.5 mol/L HNO3 and 0.5 mol/L NaOH: 1.5, 2, 3, 4, 5, 6, 7, 8, 9, 10, and 11."

Was the pH adjusted after of the partition? This could change the volume and therefore the concentration of metal.

Response 5: Thanks for your comments. Yes, the pH was adjusted after partition.

Point 6: Line 147 to 151: "To assess the influence of the factors on the adsorption of the samples, an L16(44) orthogonal experiment with 4-factor, 4-level was designed [39], which is shown in Table 1. The 16 above samples were oscillated for 4 h at a rate of 300 rpm before they were centrifuged. Their chromium ion concentrations were then measured at equilibrium. The data were analyzed with SPSS statistics (IBM 150 21, USA)." 

The manuscript should explain in simple words the meaning and usefulness of this statistic analysis.

Response 6: Thanks for your comments. The meaning and usefulness of this statistic analysis has been added in the revised manuscript.

Point 7: Line 241 and 242: "Figure 4. Adsorption isotherms of the modified peat in the Langmuir (red) and Freundlich (blue) models."

The figure 4 should be placed after of the explanation of the Langmuir and Freundlich model.

Response 7: Thanks for your comments. The figure 4 has been moved in the revised manuscript.

Point 8: Line 267: Table 3. Isotherm parameters

Two points:

1) The used units for the Langmuir model  in the table for the maximum amount adsorbed predicted by the model (qmax) are mg/g . The units used in the explanation of the Langmuir model in the manuscript (Line 264) and the Figure 4 are mg/kg. What was the reason of the inconsistency in the units? What unit was used for the calculation of qmax?

Response 8: Thanks for your comments. The units has been unified in the revised manuscript.

2) How the value of qmax of Langmuir was calculated? Was a linear equation form of Langmuir used?  If the answer is yes, the linear equation should be included in the explanation of the model.

Response 8: Thanks for your comments. Langmuir model is not transformed into linear fitting, and the qmax is calculated by fitting software

Point 9: Lines 274 to 277: "According to Kołoczek et al. [32], the  adsorption capacity of Cr(VI) (adsorption capacity of 18.75 mg/g) was significantly inferior to the  values presented in this study. A research through column experiment concluded that the best Cr(VI) adsorption capacity can reach 65.87 mg/g, which currently cannot be achieved with natural peat [33]."

These sentences should be rewritten because they are confusing, especially the last one.

Response 9: Thanks for your comments. This sentence has been changed in the revised manuscript.

Point 10: Line 280: "The adsorption properties were studied in a dynamic regime."

What is the meaning of this sentence?

Response 10: Thanks for your comments. This sentence has been deleted in the revised manuscript to avoid confusion.

Point 11: Lines 291 and 304:  "In these Equation, Qe is the equilibrium maximum adsorbance (mg/kg)"  and "According to Fig. 5A, the adsorption capacity is approximately 22mg/g"

Two points:

1) Line 291, What is the usefulness of using  mg/kg  as units instead of mg/g?  The last one is the usual unit seen in the literature and three zeros could be avoided .

Response 11: Thanks for your comments. The units has been unified in the revised manuscript.

2) Line 304, the authors are not consistent in the units.

Response 11: Thanks for your comments. The units has been unified in the revised manuscript.

Point 12: Lines 302 and 303: Table 4. Parameters of the linear forms of the pseudo-first adsorption kinetic, pseudo-second adsorption kinetic, and Elovich models.

It is not possible for k to be negative, the authors are wrong in the interpretation of the pseudo-first order adsorption kinetic equation and how to obtain the value of the rate constant.   If a reaction is of first order, a linear plot has a slope of -k.

Response 12: Thanks for your comments. The error has been changed in the revised manuscript.

Point 13: Lines 328 to 333: "Figure 6. Effects of the adsorbent dose (A), temperature (B), and pH (C) on the removal of Cr(VI) by the modified peat." 

The author concludes that the adsorption is an endothermic process based on Figure 6 B but the difference in percent of removal of metal due to the temperature between 10℃   and 50℃is not significant. Likewise there are not correlation between the increase in temperature and the increase in percent of removal of metal at some points. The percent of removal of metal falls at the temperatures of 20℃and 40℃. How these results support the idea of an endothermic process?

Response 13: Thanks for your comments. The general trend is that the removal efficiency increases with the increase of temperature, but the effect is not significant, which is also obtained in the orthogonal experiment and variance analysis.  

Point 14: Lines 346 to 349: "3C+ 2Cr2O72-+ 16H+→4Cr3++ 3CO2+ 8H2O"  

What is the meaning of 3C? The phrase "C represents the organic matter in this experiment" is a too broad definition and it is not enough to understand how the authors obtained the coefficients of the equation.   Even 3C could be seen as atomic Carbon.

Response 14: Thanks for your comments. C represents various functional groups in the organic matter in this experiment.

Point 15: Line 356 to 361

Two points:

1) This paragraph is confusing. What are the authors trying to say in the paragraph?

2) The authors recognize the possibility of precipitation at higher levels of pH but they do not mention the consequences of this precipitation in the results of the current experiment.   

Response 15: Thanks for your comments. This explanation is based on the relevant references. In solution, hexavalent chromium may be converted to trivalent chromium, which continues to react with other substances to form a precipitate that eventually removes it.

Minor points:

Point 16: Line 27 : "chrome residues are the least neglected" 

What is the meaning of the phrase?

Response 16: Thanks for your comments. This sentence has been changed in the revised manuscript.

Point 17: Line 36 and 73: "increased attentions"

The plural form is not correct.

Response 17: Thanks for your comments. This error has been changed in the revised manuscript.

Point 18: Line 74: "some modification method do not present"

The word should be "methods".

Response 18: Thanks for your comments. This sentence has been deleted in the revised manuscript.

Point 19: Line 85: The source of concentrated sulfuric acid.

Response 19: Thanks for your comments. The source of concentrated sulfuric acid has been added in the revised manuscript.

Point 20: Line 94: "whereupon the best is identified."

The word should be "was".

Response 20: Thanks for your comments. This sentence has been changed in the revised manuscript.

Point 21: Line 95: The source of potassium dichromate (K2Cr2O7).

Response 21: Thanks for your comments. The source of potassium dichromate has been added in the revised manuscript.

Round 2

Reviewer 1 Report

From the previous review, I found that the article was substantially improved, but still some corrections need to be made, namely:

Abstract

·         Line 11: Instead of “adding sulfuric acid treatments” write “adding sulfuric acid”

·         Line 12: instead of “looking for producing a modified peat with optimized Cr(VI) adsorption effect” write “with the aim of producing a modified peat with optimized Cr(VI) adsorption capability”

Materials and Methods

·         Line 84: instead of “such as 100 g peat were added to 400 mL sulfuric acid” write “(100 g peat in 400 mL sulfuric acid)”.

·         Line 85: instead of “provided” write “was provided”

·         Line 90: instead of “Acid liquor” write “Acid liquors”

·         Line 96: instead of “provided” write “was provided”

·         Lines 115, 128: dosage of adsorbent was removed, why? Your response that adsorbent dosage does not influence the results is speculative, since by Fig. 6A, later found in the text, for adsorbent dosages below 5 g/L as you have used for these experiments (per the previous version of the manuscript), the dosage is relevant.

·         Lines 148-150: dosage of adsorbent is missing.

·         Line 150: eliminate “after which single factor experiments were conducted”

·         Line 152: ref. 39 is now more appropriate, but still the design there presented contains 5 factors; how did you adapt it for 4 factors?

·         Line 156: instead of “through orthogonal experiment” write “through the orthogonal experiment”

·         Line 157: instead of “can be optimized” write “can be found”

Results and Discussion

·         Line 189: instead of “tubulous in shape, the pores of the modified one” write “tubulous in shape. The pores of the modified one”

·         Lines 189-191: instead of “were not only clearer but also filled with some other substances, which may help explain the diminishing pore volume of the modified peat”, write “seemed to be present inside larger macropores, but these seemed to be filled with smaller structures, which may explain the observed decrease in pore volume”

·         Line 191: instead of “seem to that” write “seem to show that”

·         Line 192: instead of “sulfuric acid” write “sulfuric acid treatment”

·         Table 2: the significant digits are still too many in this Table for surface area and pore size, I doubt the values are significant to the fourth decimal digit.

·         Lines 223-229: The explanation of the FTIR spectra has still many inconsistencies. Looking more deeply into FTIR interpretation, I think that you are misidentifying the 1169 cm-1 and 1035 cm-1 peaks. Where are you basing your interpretations, since no references are given? Consulting Coates (2000) “Interpretation of Infrared Spectra, A Practical Approach” (in “Encyclopedia of Analytical Chemistry”, pp.10815-10837) I’d advance that 1169 cm-1 is not C-O-C stretching vibration and is actually the sulfonate group, and not 1035 cm-1, as you referred. This would point to the sulfonate group as the responsible one for adsorption, which also makes sense given the increase in capacity with the modification. 1035 cm-1, on the other hand, could be anything from cyclohexane ring vibrations to C-C or aromatic C-H vibrations, C-O from alcohols – you are the one to better interpret this as you know your peat sample better. But it should correspond to some chemical linkage that is partly destroyed by the sulfuric acid treatment, since this band is decreased (and not increased, as you mention) with the sulfuric acid modification.

·         Table 3: refer to KF units

·         Line 267: correct significant digits to be consistent through the text, to 105.4 mg/g

·         Lines 268-270: instead of “batch adsorption experiments were performed for the removal of Cr(VI) ions from aqueous solutions using Canadian peat, the adsorption capacity of Cr(VI) (adsorption capacity of 18.75 mg/g) was significantly inferior to the values presented in this study.”, write “the adsorption capacity of Cr(VI) in Canadian peat (reported as 18.75 mg/g) was significantly inferior to the values presented in this study.”

·         Lines 270-273: Instead of “A research through column experiment removed of hexavalent chromium using sphagnum moss peat and concluded that the best Cr(VI) adsorption capacity can reach 65.87 mg/g, which currently cannot be achieved with natural peat [33].”, write “A research through column experiment using sphagnum moss peat achieved a Cr(VI) adsorption capacity of 65.87 mg/g, which currently cannot be achieved with natural peat [33].”

·         Line 273: instead of “In addition, by comparing”, write “Therefore, comparing”

·         Lines 277-278: instead of “The charts suggest that the adsorb proceeds take approximately 4 h until adsorbing capacity equilibrium is reached”, write “The charts suggest that the adsorption proceeds for 4 h, until the adsorption equilibrium is reached”

·         Line 292: α and β coefficients have meaning and units, as follows : α is the initial adsorption rate (mg g-1 h-1), and β is the reciprocal of the surface coverage when the adsorption rate is 1/e of its initial value (g mg-1) – see Wang, Z., Ainsworth, C.C., Friedrich, D.M., Gassman, P.L., Joly, A.G., 2000. Kinetics and mechanism of surface reaction of salicylate on alumina in colloidal aqueous suspension. Geochim. Cosmochim. Acta 64, 1159–1172.

·         Line 297: instead of “Adsorption kinetics with (A),” write “Adsorption kinetics (A), with”

·         Table 4: you are presenting only the slope and correlation coefficient. From the slope and intercept (shown also in Fig. 3) you can derive the parameters (qe, k1, k2, α, β) which are what is truly relevant. Otherwise this table makes no sense and you could present the r2 values in the graphs of Fig. 3.

·         Line 308-309: confusing sentence, I suggest removing.

·         Line 318: actually, by Fig. 6A, the removal efficiency seems to be increasing for dosages even above 5 g/L, please check the value of 3.2 g/L as the plateau.

·         You have not revised the explanation given in lines 347-352. By your explanation, chromium removal should increase with pH. The new sentence added in lines 336-337 also seems to support this conclusion; however, the experimental results in Fig. 6 suggest otherwise, they show that removal efficiency decreases with increasing pH. Therefore, either your explanation or the results are not valid. Assuming that you do not have any strong experimental errors that would cause a complete modification of the experimental results, you need to change lines 336-337 and 347-352 to provide an interpretation of the chemical reactions that is consistent and explains the results that you show here.

·         Lines 356-357: clarify “the poor value of K”, what does this mean?

·         Line 358: instead of “test results is” write “test results”

·         Table 5: please justify why the low level of pH is 3, since in Fig. 6A pH 2 presents the highest removal efficiency value.

·         Lines 373-374: instead of “effect whereas almost equal effect in the initial concentration, temperature, and peat dose generated similar factor effect levels.”, write “removal efficiency whereas almost equal effect in efficiency is generated by initial concentration, temperature and peat dose.”

Conclusions

·         Line 380: instead of “105.404 mg/g”, write “105.4 mg/g”

In particular, the authors absolutely need to address the scientific discussion given for FTIR interpretation and variation of removal efficiency with pH, as these are my main concerns with the paper.

I hope you find my comments of interest for the improvement of the manuscript.

Author Response

Response to Reviewer 1 Comments

Abstract

Point 1: Line 11: Instead of “adding sulfuric acid treatments” write “adding sulfuric acid”

Response 1: Thanks for your comments. "adding sulfuric acid treatments" has been changed to "adding sulfuric acid" in the revised manuscript.

Point 2: Line 12: instead of “looking for producing a modified peat with optimized Cr(VI) adsorption effect” write “with the aim of producing a modified peat with optimized Cr(VI) adsorption capability”

Response 2: Thanks for your comments. "looking for producing a modified peat with optimized Cr(VI) adsorption effect" has been changed to "with the aim of producing a modified peat with optimized Cr(VI) adsorption capability" in the revised manuscript.

Materials and Methods

Point 3: Line 84: instead of “such as 100 g peat were added to 400 mL sulfuric acid” write “(100 g peat in 400 mL sulfuric acid)”.

Response 3: Thanks for your comments. "such as 100 g peat were added to 400 mL sulfuric acid" has been changed to "(100 g peat in 400 mL sulfuric acid)" in the revised manuscript.

Point 4: Line 85: instead of “provided” write “was provided”

Response 4: Thanks for your comments. "provided" has been changed to "was provided" in the revised manuscript.

Point 5: Line 90: instead of “Acid liquor” write “Acid liquors”

Response 5: Thanks for your comments. "Acid liquor" has been changed to "Acid liquors" in the revised manuscript.

Point 6: Line 96: instead of “provided” write “was provided”

Response 6: Thanks for your comments. "provided" has been changed to "was provided" in the revised manuscript.

Point 7: Lines 115, 128: dosage of adsorbent was removed, why? Your response that adsorbent dosage does not influence the results is speculative, since by Fig. 6A, later found in the text, for adsorbent dosages below 5 g/L as you have used for these experiments (per the previous version of the manuscript), the dosage is relevant.

Response 7: Thanks for your comments. What I mean is that the adsorbent does not affect the isothermal and kinetic experiments, but affect the adsorption efficiency. But as you comments, it doesn't seem appropriate to delete it. In fact, we have done many isothermal experiments with different addition amounts (0.25g, 0.1g, and so on) of peat, but we were worried that the adsorption equilibrium state could not be reached, so we chose a lower addition amount for the fitting.

Point 8: Lines 148-150: dosage of adsorbent is missing.

Response 8: Thanks for your comments. The dosage of adsorbent has been added in the revised manuscript.

Point 9: Line 150: eliminate “after which single factor experiments were conducted”

Response 9: Thanks for your comments. This sentence has been deleted in the revised manuscript.

Point 10: Line 152: ref. 39 is now more appropriate, but still the design there presented contains 5 factors; how did you adapt it for 4 factors?

Response 10: Thanks for your comments. We designed four influencing factors, there may be more influencing factors, which is what we will continue to study in the future. Only four influencing factors are studied here, which are designed according to the orthogonal experimental table of four factors and four levels.  

Point 11: Line 156: instead of “through orthogonal experiment” write “through the orthogonal experiment”

Response 11: Thanks for your comments. "through orthogonal experiment" has been changed to "through the orthogonal experiment" in the revised manuscript.

Point 12: Line 157: instead of “can be optimized” write “can be found”

Response 12: Thanks for your comments. "can be optimized" has been changed to "can be found" in the revised manuscript.

Results and Discussion

Point 13: Line 189: instead of “tubulous in shape, the pores of the modified one” write “tubulous in shape. The pores of the modified one”

Response 13: Thanks for your comments. "tubulous in shape, the pores of the modified one" has been changed to "tubulous in shape. The pores of the modified one" in the revised manuscript.

Point 14: Lines 189-191: instead of “were not only clearer but also filled with some other substances, which may help explain the diminishing pore volume of the modified peat”, write “seemed to be present inside larger macropores, but these seemed to be filled with smaller structures, which may explain the observed decrease in pore volume”

Response 14: Thanks for your comments. "were not only clearer but also filled with some other substances, which may help explain the diminishing pore volume of the modified peat" has been changed to "seemed to be present inside larger macropores, but these seemed to be filled with smaller structures, which may explain the observed decrease in pore volume" in the revised manuscript.

Point 15: Line 191: instead of “seem to that” write “seem to show that”

Response 15: Thanks for your comments. "seem to that" has been changed to "seem to show that" in the revised manuscript.

Point 16: Line 192: instead of “sulfuric acid” write “sulfuric acid treatment”

Response 16: Thanks for your comments. "sulfuric acid" has been changed to "sulfuric acid treatment" in the revised manuscript.

Point 17: Table 2: the significant digits are still too many in this Table for surface area and pore size, I doubt the values are significant to the fourth decimal digit.

Response 17: Thanks for your comments. Significant digits has been changed in the revised manuscript.

Point 18: Lines 223-229: The explanation of the FTIR spectra has still many inconsistencies. Looking more deeply into FTIR interpretation, I think that you are misidentifying the 1169 cm-1 and 1035 cm-1 peaks. Where are you basing your interpretations, since no references are given? Consulting Coates (2000) “Interpretation of Infrared Spectra, A Practical Approach” (in “Encyclopedia of Analytical Chemistry”, pp.10815-10837) I’d advance that 1169 cm-1 is not C-O-C stretching vibration and is actually the sulfonate group, and not 1035 cm-1, as you referred. This would point to the sulfonate group as the responsible one for adsorption, which also makes sense given the increase in capacity with the modification. 1035 cm-1, on the other hand, could be anything from cyclohexane ring vibrations to C-C or aromatic C-H vibrations, C-O from alcohols – you are the one to better interpret this as you know your peat sample better. But it should correspond to some chemical linkage that is partly destroyed by the sulfuric acid treatment, since this band is decreased (and not increased, as you mention) with the sulfuric acid modification.

Response 18: Thanks for your comments. Your advice made perfect sense to me. 1169 cm-1 is actually the sulfonate group. But 1035 cm-1 is also a sulfonate group. Here we added new references. And we changed the wrong analysis in the revised manuscript.

Jiang, Z.; Zhao, X.; Manthiram, A., Sulfonated poly(ether ether ketone) membranes with sulfonated graphene oxide fillers for direct methanol fuel cells. International Journal of Hydrogen Energy 2013, 38, 5875-5884.

Point 19: Table 3: refer to KF units

Response 19: Thanks for your comments. KF units has been added in the revised manuscript.

Point 20: Line 267: correct significant digits to be consistent through the text, to 105.4 mg/g

Response 20: Thanks for your comments. The mistake has been changed through the text in the revised manuscript.

Point 21: Lines 268-270: instead of “batch adsorption experiments were performed for the removal of Cr(VI) ions from aqueous solutions using Canadian peat, the adsorption capacity of Cr(VI) (adsorption capacity of 18.75 mg/g) was significantly inferior to the values presented in this study.”, write “the adsorption capacity of Cr(VI) in Canadian peat (reported as 18.75 mg/g) was significantly inferior to the values presented in this study.”

Response 21: Thanks for your comments. This sentence has been changed in the revised manuscript.

Point 22: Lines 270-273: Instead of “A research through column experiment removed of hexavalent chromium using sphagnum moss peat and concluded that the best Cr(VI) adsorption capacity can reach 65.87 mg/g, which currently cannot be achieved with natural peat [33].”, write “A research through column experiment using sphagnum moss peat achieved a Cr(VI) adsorption capacity of 65.87 mg/g, which currently cannot be achieved with natural peat [33].”

Response 22: Thanks for your comments. This sentence has been changed in the revised manuscript.

Point 23: Line 273: instead of “In addition, by comparing”, write “Therefore, comparing”

Response 23: Thanks for your comments. "In addition, by comparing" has been changed to "Therefore, comparing" in the revised manuscript.

Point 24: Lines 277-278: instead of “The charts suggest that the adsorb proceeds take approximately 4 h until adsorbing capacity equilibrium is reached”, write “The charts suggest that the adsorption proceeds for 4 h, until the adsorption equilibrium is reached”

Response 24: Thanks for your comments. "The charts suggest that the adsorb proceeds take approximately 4 h until adsorbing capacity equilibrium is reached" has been changed to "The charts suggest that the adsorption proceeds for 4 h, until the adsorption equilibrium is reached" in the revised manuscript.

Point 25: Line 292: α and β coefficients have meaning and units, as follows : α is the initial adsorption rate (mg g-1 h-1), and β is the reciprocal of the surface coverage when the adsorption rate is 1/e of its initial value (g mg-1) – see Wang, Z., Ainsworth, C.C., Friedrich, D.M., Gassman, P.L., Joly, A.G., 2000. Kinetics and mechanism of surface reaction of salicylate on alumina in colloidal aqueous suspension. Geochim. Cosmochim. Acta 64, 1159–1172.

Response 25: Thanks for your comments. The meaning of α and β have been added in the revised manuscript. This reference is also cited.

Point 26: Line 297: instead of “Adsorption kinetics with (A),” write “Adsorption kinetics (A), with”

Response 26: Thanks for your comments. "Adsorption kinetics with (A)" has been changed to "Adsorption kinetics (A), with" in the revised manuscript.

Point 27: Table 4: you are presenting only the slope and correlation coefficient. From the slope and intercept (shown also in Fig. 3) you can derive the parameters (qe, k1, k2, α, β) which are what is truly relevant. Otherwise this table makes no sense and you could present the r2 values in the graphs of Fig. 3.

Response 27: Thanks for your comments. Here, we only want to know the R-Squared through the linear fitting, and we haven't given much consideration to the other parameters. Your suggestions have important reference significance for our following research. But here we choose to delete the table that makes no sense.

Point 28: Line 308-309: confusing sentence, I suggest removing.

Response 28: Thanks for your comments. This sentence has been deleted in the revised manuscript.

Point 29: Line 318: actually, by Fig. 6A, the removal efficiency seems to be increasing for dosages even above 5 g/L, please check the value of 3.2 g/L as the plateau.

Response 29: Thanks for your comments. This mistake has been changed in the revised manuscript.

Point 30: You have not revised the explanation given in lines 347-352. By your explanation, chromium removal should increase with pH. The new sentence added in lines 336-337 also seems to support this conclusion; however, the experimental results in Fig. 6 suggest otherwise, they show that removal efficiency decreases with increasing pH. Therefore, either your explanation or the results are not valid. Assuming that you do not have any strong experimental errors that would cause a complete modification of the experimental results, you need to change lines 336-337 and 347-352 to provide an interpretation of the chemical reactions that is consistent and explains the results that you show here.

Response 30: Thanks for your comments. After repeated experiments, the results are correct, but the analysis may lead to misunderstanding. The mechanism analysis is based on the references and chemical reactions. For this part of the analysis, we have adjusted in the revised manuscript.

Point 31: Lines 356-357: clarify “the poor value of K”, what does this mean?

Response 31: Thanks for your comments. This mistake has been changed in the revised manuscript. K is range value

Point 32: Line 358: instead of “test results is” write “test results”

Response 32: Thanks for your comments. "test results is" has been changed to "test results" in the revised manuscript.

Point 33: Table 5: please justify why the low level of pH is 3, since in Fig. 6A pH 2 presents the highest removal efficiency value.

Response 33: Thanks for your comments. In the process of adjusting pH, we found that it takes a lot of acid to adjust pH below 3, so we chose such a concentration gradient considering the actual situation.

Point 34: Lines 373-374: instead of “effect whereas almost equal effect in the initial concentration, temperature, and peat dose generated similar factor effect levels.”, write “removal efficiency whereas almost equal effect in efficiency is generated by initial concentration, temperature and peat dose.”

Response 34: Thanks for your comments. "effect whereas almost equal effect in the initial concentration, temperature, and peat dose generated similar factor effect levels" has been changed to "removal efficiency whereas almost equal effect in efficiency is generated by initial concentration, temperature and peat dose" in the revised manuscript.

Conclusions

Point 35: Line 380: instead of “105.404 mg/g”, write “105.4 mg/g”

Response 35: Thanks for your comments. This mistake has been changed in the revised manuscript.

Round 3

Reviewer 1 Report

I am sorry but I cannot recommend this article for publication while the scientific inconsistency of the explanation for Cr removal with pH is still present. The results show decrease of Cr adsorption with pH; in the text and with reactions, you describe that Cr removal should increase with pH.

Therefore, your discussion contradicts your results.

I hope you can reflect and improve this issue with your paper.